J Physiol 603.8 (2025) pp 2443–2463

# Intrinsic properties of spinal motoneurons degrade ankle torque control in humans

James A. Beauchamp[1,2] 🆔, Gregory E. P. Pearcey[2,3] 🆔, Obaid U. Khurram[4] 🆔, Francesco Negro[5] 🆔, Julius P. A. Dewald[1,2,6] 🆔 and C. J. Heckman[2,6,7,8] 🆔

[1]*Department of Biomedical Engineering, McCormick School of Engineering, Northwestern University, Chicago, IL, USA*

[2]*Department of Physical Therapy and Human Movement Sciences, Feinberg School of Medicine, Northwestern University, Chicago, IL, USA*

[3]*School of Human Kinetics and Recreation, Memorial University of Newfoundland, St. John's, NL, Canada*

[4]*Department of Physiology and Biomedical Engineering, Mayo Clinic, Rochester, MN, USA*

[5]*Department of Clinical and Experimental Sciences, Universita degli Studi di Brescia, Brescia, Italy*

[6]*Department of Physical Medicine and Rehabilitation, Feinberg School of Medicine, Northwestern University, Chicago, IL, USA*

[7]*Department of Neuroscience, Feinberg School of Medicine, Northwestern University, Chicago, IL, USA*

[8]*Shirley Ryan AbilityLab, Chicago, IL, USA*

Handling Editors: Richard Carson & Madeleine Lowery

The peer review history is available in the Supporting Information section of this article (https://doi.org/10.1113/JP287446#support-information-section).

**Abstract figure legend** All motor commands are processed via spinal motoneurons, whose intrinsic electrical properties are adapted by brainstem neuromodulatory inputs. The effects of these neuromodulatory inputs (i.e. persistent inward currents; PICs) must be precisely regulated by inhibitory mechanisms to support the full range of human motor behaviours. Here, we introduce the sombrero contraction (1), an isometric task consisting of a linear ramp atop a stabilizing hold that is designed to challenge this inhibitory control. In this task, upon returning to the second hold phase, the inhibitory input required to deactivate PICs in motor units recruited during the ramp phase is compromised,

This article was first published as a preprint. Beauchamp JA, Pearcey GEP, Khurram OU, Negro F, Dewald JPA, Heckman CJ. 2023. Intrinsic properties of spinal motoneurons degrade ankle torque control in humans. bioRxiv. https://doi.org/10.1101/2023.10.23.563670

The Journal of Physiology

as sustained excitatory drive is needed to maintain the hold. As a result, many of these newly recruited motor units continue firing, leading to increased torque variability and difficulty in control. To further illustrate these challenges in controlling PICs, we modulated the balance of excitatory and inhibitory input to the motor pool by altering muscle length before isometric ramp contractions, altering the inhibition available for PIC deactivation. We show that PICs are greatest when a muscle is put in a relatively shorter length (2), and that this change amplifies the challenges in controlling PICs identified in the sombrero contraction (3). (MVT, maximum voluntary torque; PPS, pulse-per-second).

**Abstract**  Motoneurons are the final common pathway for all motor commands and possess intrinsic electrical properties that must be tuned to control muscle across the full range of motor behaviours. Neuromodulatory input from the brainstem is probably essential for adapting motoneuron properties to match this diversity of motor tasks. A primary mechanism of this adaptation, control of dendritic persistent inward currents (PICs) in motoneurons by brainstem monoaminergic systems, generates both amplification and prolongation of synaptic inputs. While essential, there is an inherent tension between this amplification and prolongation. Although amplification by PICs allows for quick recruitment and acceleration of motoneuron discharge, PICs must be deactivated to derecruit motoneurons upon movement cessation. In contrast, during stabilizing or postural tasks, PIC-induced prolongation of synaptic inputs is critical for sustained motoneuron discharge. Here, we designed two motor tasks that challenged the inhibitory control of PICs, generating unduly PIC prolongation that increases variability in human torque control. This included a paradigm combining a discrete motor task with a stabilizing task and another involving muscle length-induced changes to the balance of excitatory and inhibitory inputs available for controlling PICs. We show that prolongation from PICs introduces difficulties in ankle torque control and that these difficulties are further degraded at shorter muscle lengths when PIC prolongation is greatest. These results highlight the necessity for inhibitory control of PICs and showcase issues introduced when inhibitory control is constrained. Our findings suggest that, like sensory systems, errors are inherent in motor systems. These errors are not due to problems in the perception of movement-related sensory input but are embedded in the final stage of motor output. This has many implications relevant to clinical conditions (e.g. chronic stroke) where pathological shifts in mono-amines may further amplify these errors.

(Received 5 August 2024; accepted after revision 26 February 2025; first published online 28 March 2025)

**Corresponding author** C. J. Heckman: Department of Neuroscience, Feinberg School of Medicine, Northwestern University, Chicago, IL 60611, USA.    Email: c-heckman@northwestern.edu

### Key points

- All motor commands are processed via spinal motoneurons, whose intrinsic electrical properties are adapted by brainstem neuromodulatory input.
- The effects of these neuromodulatory inputs (i.e. persistent inward currents; PICs) must be tightly regulated by inhibitory inputs to allow for the large repertoire of human motor behaviours.
- We designed two motor tasks to restrict the ability of inhibitory synaptic inputs to control PICs and show that this generates substantial errors that reduce the precision of motor output in humans.
- Our findings suggest that errors are inherent in motor systems and embedded in the final stage of motor output. This has many implications relevant to clinical conditions (e.g. chronic stroke) and may, speculatively, shed light on contributing factors to muscle cramps.

## Introduction

Alpha motoneurons are the final common output of the central nervous system and transduce all motor commands (i.e. synaptic inputs) into discharge patterns for the generation of force. Thus, it is reasonable to assume that the intrinsic electrical properties that govern this transduction are adapted to facilitate the large repertoire of motor behaviours observed in humans (Johnson et al., 2017). However, insight into how motoneurons effectively

implement motor commands has been complicated by the discovery that their intrinsic electrical properties are profoundly altered by brainstem-originating neuromodulatory inputs (e.g. noradrenaline, serotonin) (Hounsgaard et al., 1984; Lee & Heckman, 1998; Schwindt & Crill, 1977, 1980).

Brainstem axons that release the monoamines serotonin (5HT) or noradrenaline (NA) form dense, monosynaptic projections to the spinal cord and provide the primary control of motoneuron excitability (Alvarez et al., 1998; Bowker et al., 1982; Holstege & Kuypers, 1987). In adult motoneurons, 5HT and NA facilitate voltage-sensitive sodium and calcium channels that mediate persistent inward currents (PICs) (Hounsgaard & Kiehn, 1985; Hounsgaard et al., 1988; Lee & Heckman, 1999). PICs provide an additional source of depolarizing (i.e. excitatory) current that both amplifies and prolongs synaptic inputs to motoneurons by as much as 3–5-fold, imparting distinct characteristics to their discharge patterns (Binder et al., 2020; Lee & Heckman, 1996, 2000). These distinct characteristics are clearly manifest in human motor unit discharge patterns and are likely fundamental for normal motor output (Heckman & Enoka, 2012; Khurram et al., 2022).

Brainstem neuromodulation of PICs by monoamines, and the resulting amplification and prolongation of excitatory synaptic inputs, have multiple important functions. Amplification from PICs augments the modest synaptic currents generated by individual excitatory input systems so that each can strongly affect motoneuron discharge (Binder & Powers, 2001; Cushing et al., 2005). Likewise, in parallel, the prolongation of discharge provided by PICs is probably essential for postural tasks and underlies the self-sustained discharge of motoneurons that is observed following brief excitatory inputs in intracellular recordings (i.e. bistable behaviour) (Crone et al., 1988; Heckman, Johnson et al., 2008; Hounsgaard et al., 1988). Consequently, modulation of PICs provides the basis for gain control of motor output, by which monoamines may adjust motoneuron excitability to match the widely varying force demands of normal human movement (Johnson & Heckman, 2014; Naufel et al., 2019; Wei et al., 2014).

While essential, there is potential for conflict between the amplification and prolongation of excitatory inputs provided by PICs. Though amplification by PICs allows for quick recruitment and acceleration of motoneuron discharge, these PICs must be deactivated to derecruit motoneurons and avoid sustained activation upon movement cessation. In contrast, continued activation of PICs (i.e. prolongation) is probably necessary for sustained contractions in stabilization and postural tasks. These contrasting task-dependent demands necessitate a precise control mechanism to adjust their magnitude and duration. While monoaminergic drive adjusts the magnitude of PICs, these effects lack the specificity necessary to independently adjust PIC amplification and prolongation. Thus, contrasting task demands are likely accomplished by precise inhibitory control embedded within motor commands (Bui et al., 2008; Heckman, Hyngstrom et al., 2008; Hyngstrom et al., 2007; Johnson et al., 2012; Kuo et al., 2003). Therefore, we reasoned that tasks that challenge the capacity of inhibitory inputs to regulate PIC behaviours would lead to aberrant motoneuron discharge patterns and poorly controlled forces.

We designed two isometric contraction paradigms that were likely to constrain inhibitory control of PICs. In the first, we superimposed a discrete motor task and a prolonged low-effort stabilization task. In this superimposition task (colloquially termed sombrero), the inhibitory input necessary to deactivate the PICs of higher threshold motoneurons recruited during the discrete task is compromised because of the sustained excitatory input required for the stabilizing task. Therefore, we hypothesized that PIC prolongation of higher threshold motor units, otherwise not required for this low-level output, would be excessive after the return to stabilization and result in highly variable torque control. In the second paradigm, we modulated the balance of excitatory and inhibitory input available for controlling PICs by changing the length of agonist–antagonist muscle pairs prior to isometric motor tasks. Studies in humans have highlighted an increase in spinal excitability as a muscle is shortened, probably yielding a relative reduction in net inhibitory input to the motor pool and hindering PIC deactivation (Dutt-Mazumder et al., 2020; Frigon et al., 2007; Hwang, 2002; Patikas et al., 2004). Thus, we hypothesized that motoneuron outputs would suffer from excessive PIC prolongation at short

**James (Drew) Beauchamp** is a postdoctoral research fellow at Carnegie Mellon University's NeuroMechatronics Laboratory with Dr Douglas Weber, where he focuses on sensory neuromodulation. Drew completed his PhD in Biomedical Engineering from Northwestern University, working with Drs C. J. Heckman and Julius Dewald. His dissertation examined motor unit behaviour to uncover how motoneurons are controlled to facilitate human movement and how motor commands are disrupted in neuropathological conditions. Currently, Drew's work centres on developing and utilizing innovative bioelectronic and mechatronic devices to investigate signalling patterns within and across the distributed neural networks underlying sensation and motor control.
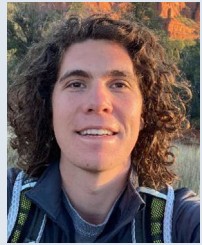

agonist muscle lengths, again resulting in poor torque control.

Our results strongly supported our hypothesis for the first paradigm and revealed excessive PIC prolongation and poor torque control following cessation of the discrete task and return to the stabilizing task. Similarly, in the second paradigm, we found greater PIC prolongation at short agonist lengths for both ankle dorsiflexors and plantar flexors. Combining these two paradigms further supported this finding and highlighted greater PIC prolongation and worsened torque control during the superimposition (sombrero) task when a muscle is in a shortened position.

Though the monoaminergic dependence of PICs allows for brainstem control of motoneuron excitability, the inherent characteristics of PICs necessitate task-dependent inhibitory control. Here we have highlighted this necessity and shown two instances where conflicts and/or constraints to inhibitory control of PICs introduce quantifiable difficulties in human torque control. These findings have many implications with relevance to clinical conditions where pathological shifts in monoamines are theorized (e.g. stroke, spinal cord injury) (Beauchamp, Urday et al., 2022; Li et al., 2019; McPherson, Ellis et al., 2018; Murray et al., 2010).

## Materials and methods

### Ethical approval

Data were collected from 12 healthy volunteers ($26.5 \pm 3.0$ years; two females, ten males) with no known history of cardiovascular, metabolic or neuromuscular impairment for each experiment. All individuals provided written and informed consent prior to participation, in accordance with Northwestern University's Institutional Review Board (STU00202964). The study conformed to the standards set by the *Declaration of Helsinki*, except for registration in a database

### Overview

The primary goal of this work was to highlight instances where the intrinsic properties of spinal motoneurons (i.e. PICs) introduce difficulties in human motor control, due to their need for task-dependent inhibitory control. To accomplish this, we designed two experimental paradigms that introduced potential constraints and conflicts to inhibitory control of PICs. We then used high-density surface electromyography (HDsEMG) to estimate human motor unit (MU) behaviour and PIC characteristics during these paradigms and observed the subsequent behavioural ramifications.

The first paradigm is presented as experiment 1, the second paradigm as experiment 2 and a combination of both paradigms as experiment 3. In the first experiment, we had individuals perform isometric plantarflexion and dorsiflexion 'sombrero' contractions, which presumably limit an individual's ability to control PICs with inhibitory inputs. In the second experiment, we modulated the balance of excitatory and inhibitory input to the motor pool by changing the length of the muscle prior to isometric ramp contractions, presumably altering the relative amount of inhibition available for PIC deactivation. In the final experiment, we had individuals perform the sombrero contractions at varying muscle lengths, utilizing the excitability changes from the second experiment to amplify the challenges in controlling PICs identified in the first experiment.

### Experimental setup

For each paradigm, similar configurations were employed. In all sessions, participants were seated in a Biodex chair, and their left foot was securely attached to a footplate fixed onto a Systems 4 Dynamometer (Biodex Medical Systems, Shirley, NY, USA) with thigh and shoulder straps employed to minimize movement. Throughout each session, the centre of rotation of a participant's ankle joint was aligned with the dynamometer's axis of rotation and their left knee and hips were maintained at 20° and 80° of flexion, respectively. In experiment 1, a participant's ankle was placed at 90° and was adjusted $\pm 20°$ in experiments 2 and 3. Target isometric contractions and visual feedback (i.e. ankle dorsiflexion or plantarflexion torque) were provided on a flat monitor via a custom Matlab interface [MATLAB (R2020b), The Mathworks Inc., Natick, MA, USA]. Torque about the ankle was filtered with a 125 ms moving average window before being provided as visual feedback to the participant. For subsequent analysis, raw torque signals were digitized (2048 Hz) using a 16-bit analog-to-digital (A/D) converter (Quattrocento, OT Bioelettronica, Turin, Italy) and lowpass filtered (50 Hz; fifth-order Butterworth filter).

### Experimental protocols

Experiments 1, 2, and 3 were conducted on the same day in random order for each participant. To normalize subsequent efforts, each experimental session began with determining a participant's maximal isometric torque-generating capacity for both dorsiflexion and plantarflexion. This was done independently for each of the three ankle angles to be tested, with the ankle positioned at 70°, 90° or 110°. For each ankle angle and torque direction, at least two maximal contractions were performed, with at least 1 min of rest separating contractions. Maximal contractions were repeated until the peak

torque within the last contraction deviated by less than 10% of the previous two contractions. We then used the maximum voluntary torque (MVT) achieved during these efforts to normalize all subsequent contractions. Prior to each paradigm, practice trials were performed until an individual could match the desired torque trajectories with less than ±5% error for more than half the trial, with real-time visual feedback of an individual's performed torque and desired torque provided. During each experiment, all trials were included unless the participant was unable to achieve the desired torque trace, ceased the trial prematurely or substantially deviated from the torque trace ±5%. Contractions were carried out in the dorsiflexion and plantarflexion direction with HDsEMG recorded from surface electrode arrays atop the skin overlying the tibialis anterior (TA) and medial gastrocnemius (MG) muscle bellies. Throughout all trials, antagonist EMG was monitored, and participants were coached to use only agonist muscles for the intended dorsiflexion or plantarflexion contraction.

**Experiment 1.** In the first paradigm, we constrained the inhibitory input available for deactivating PICs by requiring individuals to maintain excitatory drive to the motor pool in a contraction that we call a 'sombrero'. In short, individuals produced a linear increase and decrease in isometric ankle torque (i.e. ramp contraction) while performing a low-effort stabilizing isometric ankle torque task. Practically, for each trial, individuals were first asked to generate isometric dorsiflexion or plantarflexion ankle torque to 10% MVT and maintain this effort for 10 s, as precisely and accurately as possible (referred to as *plateau one*). Following 10 s, individuals were then asked to linearly increase and decrease their effort by ~3% MVT/s to 30% MVT and back to 10% MVT (referred to as *ramp*). Following their return to 10% MVT, individuals were again asked to hold 10% MVT for 10 s, as precisely and accurately as possible (referred to as *plateau two*). Control trials were randomly interspersed, in which individuals were required to maintain 10% MVT for the duration of the contraction (i.e. sustained hold). A minimum of four sustained hold and sombrero contractions were performed.

During sombrero contractions, the additional excitatory synaptic input necessary to recruit higher threshold MUs and perform the linear ramp contraction also activates the PICs in these recruited units. Furthermore, by requiring that individuals slowly relax to a maintained low-level effort (i.e. plateau two; 10% MVT hold), we are forcing a maintained base level of excitatory drive to the motor pool. This excitatory drive is in conflict with the inhibitory input that would be necessary to inactivate PICs, allowing for PICs in MUs recruited in

the ramp contraction to facilitate discharge well into the second plateau.

**Experiment 2.** In experiment 2, we modulated the balance of excitatory and inhibitory input to the motor pool by changing agonist muscle length. This was accomplished by modulating the angle of the ankle prior to isometric dorsiflexion and plantarflexion ramp tasks. Prior studies have shown an increase in spinal excitability as a muscle is shortened (Dutt-Mazumder et al., 2020; Frigon et al., 2007; Hwang, 2002). Increasing spinal excitability yields a relative reduction of inhibitory input and/or disinhibition to motoneurons, which would reduce inhibitory constraints on PICs. Prior work from our group has highlighted the unique sensitivity of PICs to inhibitory inputs and even shown ankle joint angle in the decerebrate cat to modulate PIC magnitude (Heckman, Hyngstrom et al., 2008; Hyngstrom et al., 2007; Kuo et al., 2003). Thus, we reasoned that changing the length of agonist muscles should either constrain (long muscle, reduced spinal excitability) or facilitate (short muscle, increased spinal excitability) PICs.

For all trials, individuals were asked to produce linear isometric dorsiflexion and plantarflexion ramp contractions to 30% MVT with a rise and decay speed of 3% MVT/s. This was done in random order for three separate ankle angles (70°, 90° and 110°). For each ankle angle, a minimum of four ramp contractions were performed, two each for plantarflexion or dorsiflexion. For this experiment, we performed linear isometric ramp contractions, as these contractions have well validated metrics for estimating PICs in humans (Afsharipour et al., 2020; Beauchamp, Pearcey et al., 2023; Chardon et al., 2023; Gorassini et al., 2002). All trials were adjusted for the maximum voluntary torque generated in the exact configuration to be tested.

**Experiment 3.** In experiment 3, we asked individuals to perform sombrero tasks or sustained holds at the two extreme ankle angles of experiment 2 (i.e. 70° and 90°). A minimum of 16 contractions were performed: two replicates for plantarflexion or dorsiflexion at both ankle angles for the sombrero contractions and sustained holds. As before, the desired torque trajectories supplied to participants were normalized to the maximum voluntary dorsiflexion or plantarflexion torque for a given ankle angle.

### Data analysis

Specific details regarding analysis and statistical methodologies can be found below. Briefly, HDsEMG was decomposed into individual MU spike trains using convolutive blind source separation and successive sparse

deflation improvements (Martinez-Valdes et al., 2017; Negro et al., 2016) and fit with support vector regression (Beauchamp, Khurram et al., 2022). MU discharge profiles were analysed as detailed below.

## Motor unit decomposition

HDsEMG was collected with 64-channel electrode grids (GR08MM1305, OT Bioelettronica) placed atop the skin overlying the TA and MG muscle bellies with adhesive foam and conductive paste. HDsEMG was acquired with differential amplification (150×), bandpass filtered (10–900 Hz) and digitized (2048 Hz) using a 16-bit A/D converter (Quattrocento, OT Bioelettronica).

Following collection, each channel of surface EMG was visually inspected to remove channels with substantial artifacts, noise or saturation of the A/D board. The remaining EMG channels were decomposed into individual MU spike trains using convolutive blind source separation and successive sparse deflation improvements (Martinez-Valdes et al., 2017; Negro et al., 2016). The silhouette threshold for decomposition was set to 0.87. To improve decomposition accuracy, automatic decomposition results were manually augmented by iteratively re-estimating the spike train and correcting for missed spikes or substantial deviations in the discharge profile (del Vecchio et al., 2020; Hug et al., 2021; Martinez-Valdes & Negro, 2023).

## Motor unit analysis

Following decomposition, binary MU spike trains were used to generate discrete estimates of instantaneous discharge rate. For each MU, instantaneous estimates were then smoothed with support vector regression to create continuous estimates, as previously described (Beauchamp, Khurram et al., 2022). In brief, the reciprocal of the inter-spike intervals (ISIs), or the time between consecutive spikes, were used to train a support vector regression (SVR) model with L1 soft-margin minimization to predict instantaneous discharge rate as a function of the corresponding time instances for each MU. Smooth and continuous estimates of discharge rate were then generated with this SVR model along a time vector from MU recruitment to derecruitment sampled at 2048 Hz [MATLAB (R2020b), the Mathworks Inc.]. Hyperparameters were chosen in accordance with those previously suggested (Beauchamp, Khurram et al., 2022; Beauchamp, Pearcey et al., 2023).

## Estimates of PIC-induced prolongation of discharge

For the sombrero contractions in experiments 1 and 3, estimates of PIC prolongation were facilitated by separating MUs into three cohorts based upon the torque at which they were recruited during the contraction. This included brim MUs that were recruited at the onset of the stabilization task in *plateau one*, button MUs that were recruited and derecruited in the center ramp region, and cap MUs that were recruited in the centre ramp but sustained discharge into the second plateau. The duration of sustained discharge (i.e. prolongation) from PICs was estimated as the duration of time that a cap MU maintained discharge (i.e. ISI < 1 s) after its theoretically expected point of derecruitment. These values are calculated as the duration of time (s) that an MU continues to discharge past the instance that it should have been derecruited if its recruitment and derecruited torque were identical. That is, if an MU was recruited at 15% MVT on the ascending portion of the centre ramp, its sustained discharge would be represented by the time that it continued to fire past 15% MVT on the descending portion of the ramp. Furthermore, the proportion of cap MUs that sustained discharge was quantified per trial as the total number of cap MUs that sustained discharger greater than 2 s, divided by the total number of MUs recruited in the ramp contraction (i.e. cap + brim MUs). MU recruitment and derecruitment was estimated as the time (and torque) at which the decomposition predicted the first and last instantaneous MU discharge, respectively.

In experiment 2, we employed a collection of methods to quantify PIC prolongation. These metrics are detailed below and include a paired MU analysis technique (ΔF), the duration of discharge that an MU exhibits on the descending phase of the ramp contraction and the torque at which MUs were derecruited. An additional battery of metrics was conducted and can be found in the supplementary information for the interested reader.

**Paired motor unit analysis.** Delta frequency (ΔF) is a commonly employed and well-characterized metric used to estimate the magnitude of PICs and represents the hysteresis of a higher threshold MU with respect to the discharge rate of a lower threshold unit. ΔF for a given MU (test unit) is quantified as the change in discharge rate of a lower threshold MU (reporter unit) between the recruitment and derecruitment instance of the test unit. To account for the possible pairing of a test unit with multiple lower threshold reporter units, we represented ΔF for a given test unit as the average change in discharge rate across all reporter unit pairs. To ensure the validity of ΔF estimates, we employed multiple exclusion criteria for test–reporter unit pairs; to ensure that MU pairs likely received a common synaptic drive, we only included test unit–reporter unit pairs with rate–rate correlations of $r^2 > 0.7$ (Gorassini et al., 2002; Udina et al., 2010); to ensure full activation of the PIC in the reporter unit, we excluded any pairs with recruitment time differences <1 s

(Hassan et al., 2020; Powers et al., 2008); and to avoid saturated reporter units, we excluded test unit–reporter unit pairs in which the reporter unit discharge range was <0.5 pps while the test unit was active (Stephenson & Maluf, 2011).

**Descending discharge and derecruitment.** To provide insight into the behaviour of MUs on the descending phase of discharge, we extracted two key metrics. This included the total duration of time that an MU exhibited sustained discharge following peak torque. Quantitatively, this was calculated as the time in seconds from the occurrence of peak torque for a given trial to the time of derecruitment for each MU. To further characterize the descending phase of MU discharge we also quantified the torque at which each MU was derecruited and represent this as a percentage of maximum voluntary torque.

**Supplemental metrics.** Additional metrics not highlighted in the primary results for the second experiment, but of potential interest, can be found in the supplementary information and include torque at MU recruitment, torque at MU derecruitment, the duration of MU discharge on the ascending phase of the ramp, the duration of discharge on the descending phase of the ramp, the difference in ascending and descending duration as a function of the total duration of MU discharge (i.e. ADR: ascending–descending ratio) (Afsharipour et al., 2020; Hassan et al., 2021), and the discharge rate at MU recruitment, peak and derecruitment.

### MU matching

To observe the change in metrics employed in the second experiment for presumably the same MU across each muscle length, we identified repeated observations of the same MU at each length. Identifying the same MU at each joint angle (muscle length) allows for estimating PIC prolongation whilst removing the potential bias that may be introduced by decomposing and/or recruiting varying populations of MUs at each length. To do this, we estimated MU action potential waveforms (MUAPs) with spike-triggered averaging and computed a 2-D cross-correlation between the spatial representation of the MUAPs between ramp contraction trials (del Vecchio et al., 2019; Martinez-Valdes et al., 2017). This was repeated across all MUs in successive contractions at each muscle length, with normalized correlation values between MU pairs greater than 0.8 deemed a matched unit. Matched unit pairs across trials were given a single unique MU identifier. We then collected the values for each of the proposed metrics for matched MUs to estimate the change in MU discharge introduced by changes in

muscle length. Outcome metrics for only MUs matched at a minimum of two muscle lengths can be observed in the supplementary information. These matched data follow the same trends of the unmatched MUs displayed in the results.

### Statistical analysis

To observe task performance and MU characteristics within the sombrero and hold contractions, each trail was segmented into two regions indicating either *plateau one* or *plateau two*. These regions of times were matched between the sombrero and hold contractions. To compare changes between plateaus, for the sombreros and holds independently, we quantified the average discharge rate of MUs and coefficient of variation in torque for each plateau and fit each of these metrics with a linear mixed model that contained fixed effects of a muscle (MG, TA) and plateau region (1, 2), with a random effect of participant and covariate of trial number. To appreciate the magnitude of change between plateaus, we computed estimated marginal mean differences between plateau factor levels. To compare differences between the sombrero and hold contractions, we quantified the change in each metric from *plateau one* to *two* ($\Delta$ values) for each trial and fit a linear mixed model to these differences, comprising fixed effects of muscle (MG, TA), contraction type (sombrero, hold) and their interaction with a random effect of participant and covariate of trial number. We then computed estimated marginal means for these $\Delta$ values and quantified the effect size (Cohen's *d*) for the difference in these values between the sombrero and hold contractions.

To observe changes in the prolongation of MU discharge as a function of muscle length, we fit a linear mixed model to the battery of aforementioned metrics with fixed effects of muscle (MG, TA), length (long, mid, short) and their interaction. For the unmatched population data, we employed a random effect of participant and covariate of trial number. For the matched data, we employed a random effect of MUid (a unique identifier of independent MUs). We then computed estimated marginal means for each metric and quantified the effect size (Cohen's *d*) for the difference in these values between the long and short lengths.

To observe changes in cap MU behaviour as a function of length, we quantified the sustained duration for each cap MU and fit a linear mixed model to these values comprising fixed effects of muscle (MG, TA), length (long, short) and their interaction with random effects of participant and covariate of trail number. Similarly, for each trial we quantified the number of cap MUs that sustained discharge >2 s as a function of the button MUs, the total number of cap MUs and their average

discharge rate, and fit a linear mixed model to these values comprising fixed effects of muscle (MG, TA), length (long, short) and their interaction with random effects of participant. To observe task performance and MU characteristics within the sombrero and hold contractions, we conducted a similar protocol to experiment 1. Explicitly, we quantified the change in torque and average MU discharge rate between *plateau one* and *two* for each trial and fit with a linear mixed model with fixed effects of contraction type (sombrero, hold), muscle (MG, TA), length (long, mid, short) and their interaction with a random effect of participant and covariate of trail number. We then computed estimated marginal means for each metric and quantified the effect size (Cohen's *d*) for the difference in these values between the long and short lengths and between contraction types.

All statistical analysis was performed with R. Mixed model analysis was achieved via the lme4 (Bates et al., 2015) package and *P*-values were obtained by likelihood ratio tests of the full model with the effect in question against the model without the effect in question. For main effects, this included their subsequent interaction terms. To ensure the validity of model fitting, the assumptions of linearity and normal, homoscedastic residual distributions were inspected. Effect size and estimated marginal means were employed in pairwise *post hoc* testing and achieved with the emmeans package (Lenth, 2022). Cohen's *d* effect size is employed and represents the difference in means as a function of the standard deviation. Commonly used interpretations separate effects sizes intro three categories: small ($d = 0.2$), medium ($d = 0.5$) and large ($d = 0.8$). Estimated marginal means represent the means predicted from the statistical model for each relevant independent variable combination. They allow comparisons between the dependent variables of interest, while accounting for the appropriate fixed and/or random effects in the model. Significance was set at $\alpha = 0.05$ and pairwise and multiple comparisons were corrected using Tukey's corrections for multiple comparisons.

## Results

### Experiment 1: examining PIC prolongation and its effect on torque control using sombrero trajectories

In the first paradigm, we forced a conflict between the inhibitory input necessary to deactivate PICs and a continued excitatory drive to maintain motor output, generating prolonged MU firing and difficulties in torque control. Fig. 1*A* shows an example trial for this paradigm, with the black trace indicating ankle torque and the coloured traces showing the discharge rates of decomposed MUs. The MUs are offset along the *y*-axis based on their recruitment thresholds. As indicated in Fig. 1*A*, MUs categorized as *brim* MUs were recruited at the onset of the stabilization task in plateau one, button MUs were recruited and derecruited in the centre ramp region, and *cap* MUs were recruited in the centre ramp but sustained discharge into the second plateau. The relationship between recruitment and derecruitment for the three categories of MUs can be appreciated in Fig. 1*A* (left-hand panel). Button MUs tend to be recruited and derecruited at similar torque levels, whereas cap and brim units tend to be derecruited towards the end of the contraction irrespective of recruitment thresholds.

Across both muscles, an average of 10.35 MUs (95% confidence interval (CI): [8.00–12.63]) were recruited in the centre ramp for a given trial, with trials that possess cap MUs displaying an average of 5 MUs (95% CI: [3.83–6.54]) sustaining discharge greater than 2 s into the second plateau (Fig. 1*C*). The duration of sustained discharge past an expected theoretical derecruitment is shown in the rightmost column of Fig. 1*B* (right-hand panel) and indicates PIC prolongation. This duration was estimated as the time that an MU maintained discharge after its theoretically expected point of derecruitment, with the red vertical line indicating an MU with identical recruitment and derecruitment torque. As can be appreciated, sustained discharge is significantly greater in cap MUs than in brim/button MUs ($\chi^2(2) = 1038.2$, $P < 0.001$), with cap MU sustained discharge estimated as 10.91 s (95% CI: [9.46–12.37]) and 9.51 s (95% CI: [7.94–11.08]) for the TA and MG, respectively (Fig. 1*B*).

Interestingly, despite cap MUs actively discharging during the second plateau, torque output was close in magnitude between plateaus, as 10% MVT was demanded for both. Between plateaus, we observed an average increase of 0.28% MVT (95% CI: [0.19–0.37]). Nonetheless, task performance was significantly degraded in the second plateau, as evidenced by an increase in the coefficient of variation (CV) in ankle torque (Fig. 2, left side). We found the plateau order to be predictive of CV ($\chi^2(2) = 49.72$, $P < 0.001$), with CV estimated to increase during *plateau two* by 1.73% (95% CI: [1.21–2.24, $d = 1.31$]) for dorsiflexion trials and by 1.51% (95% CI: [1.00–2.02], $d = 1.15$) for plantarflexion trials. These increases are notable when compared to the magnitude of CV during *plateau one*, increasing by more than 1.5-fold (plantar flexion: 2.60% [2.06–3.13]; dorsiflexion: 3.17% [2.62–3.71]). This increase in variation and apparent task difficulty can additionally be appreciated in the exemplar trials shown in Figs 1*A* and 2. Despite our instructions that specified precise and accurate ankle torque, individuals were more variable in *plateau two*.

In addition to an increase in torque variability, we also found brim MUs to exhibit lower average MU discharge rates in the second plateau of the sombrero (Fig. 3). Plateau order predicted average MU discharge rate ($\chi^2(2) = 607.24$, $P < 0.001$) with MUs actively

discharging in the first plateau lowering their average discharge rate in the second plateau by an estimated 1.75 pps (95% CI: [1.01–2.49]) and 2.40 pps (95% CI: [1.73–3.08]) for the TA and MG, respectively. The change in average discharge rate from *plateau one* to *two* is shown in Fig. 3, with grey lines connecting unique MUs.

To ensure that the observed MU characteristics and torque deficits were not the results of a time-dependent

habituation or a product of spike frequency adaptation, we additionally asked individuals to perform long stabilizing hold contractions (Fig. 2, right side). As expected, we found no significant recruitment of additional MUs (e.g. cap MUs) during hold trials. Furthermore, we found the order of plateau to significantly predict CV in ankle torque ($\chi^2(2) = 18.35$, $P < 0.001$) for the hold contractions, but found these values to decrease in *plateau two*

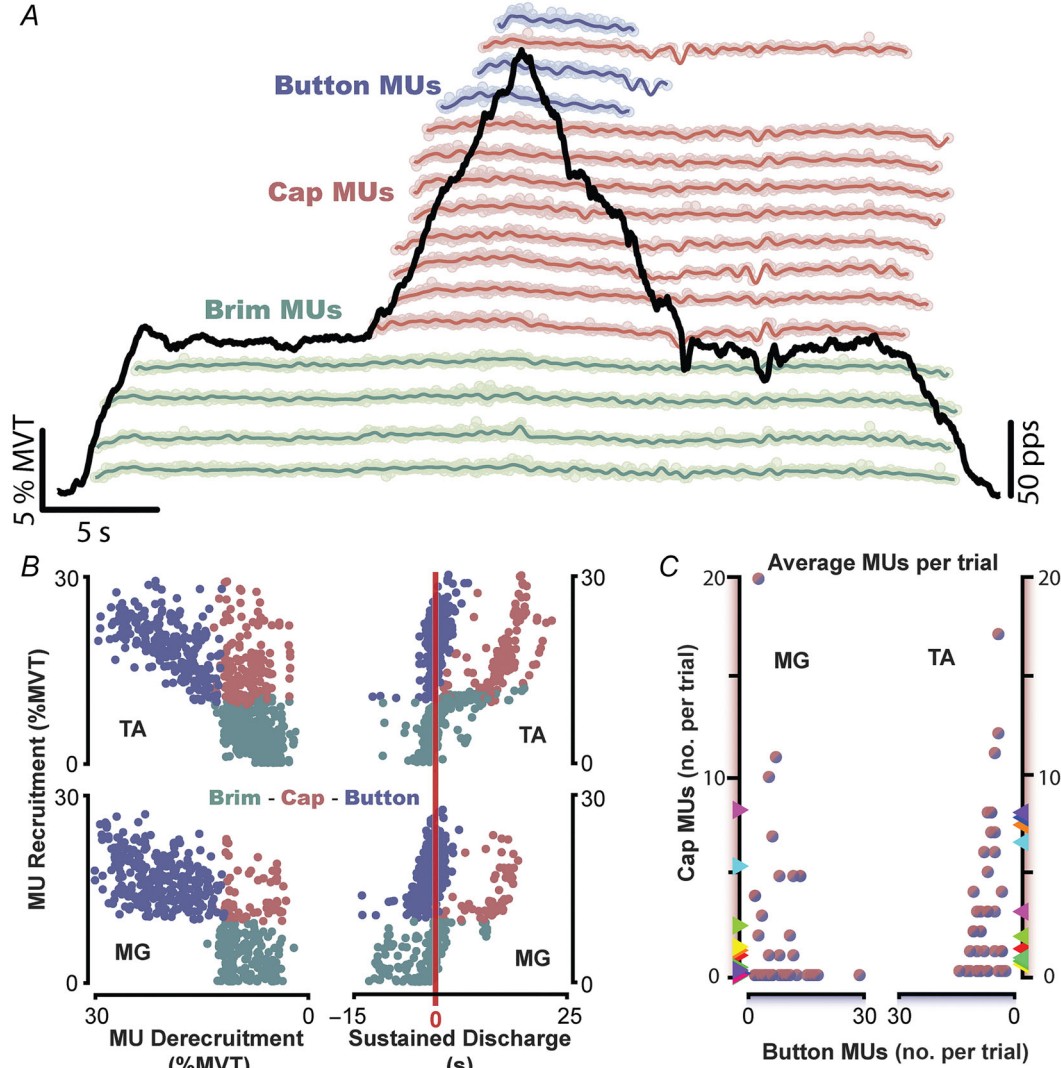

**Figure 1. The sombrero contraction: a superimposition task that constrains inhibitory control of PICs**
*A*, an exemplar sombrero contraction from a single participant, with dorsiflexion torque shown in the black trace normalized to an individual's maximum voluntary torque (MVT). A collection of decomposed motor units (MUs) is shown smoothed and offset in the *y*-axis based on when they were recruited in the contraction and coloured according to their categorization. MU categories included brim MUs that were recruited at the onset of the stabilization task in plateau one, button MUs that were recruited and derecruited in the centre ramp region, and cap MUs which were recruited in the centre ramp but sustained discharge into the second plateau. *B*, MU recruitment and derecruitment characteristics are in the left column and duration of sustained discharge on the right, with each dot representing an independent MU. Sustained discharge is represented as the duration of time past theoretical derecruitment and the vertical red line indicates 0 s (i.e. an MU with identical recruitment and derecruitment torque). *C*, the average number of cap MUs as a function of button MUs decomposed per trial. Coloured triangles along the *y*-axis represent participant averages. (PPS: pulse-per-second) [Colour figure can be viewed at wileyonlinelibrary.com]

for dorsiflexion (−1.04% [95% CI: −1.81 to −0.277], $P = 0.008$), and to not change for plantarflexion trials ($P = 0.198$). Also, we found no significant changes in MU discharge rate between *plateau one* and *two* for the TA ($P = 0.153$) and found a decrease of 1.08 pps (95% CI: [0.416–1.75], $P = 0.004$) in the MG (Fig. 3, right side).

Comparing the sombrero and hold contractions, we found that the type of task predicted both the change in torque CV (dorsiflexion: $\chi^2(1) = 19.74$, $P < 0.001$; plantarflexion: $\chi^2(1) = 11.96$, $P < 0.001$) and MU discharge rates (dorsiflexion: $\chi^2(1) = 92.80$, $P < 0.001$;

plantarflexion: $\chi^2(1) = 73.33$, $P < 0.001$). Brim MUs in the sombrero show a significantly greater decrease in discharge rate from *plateau one* to *plateau two* when compared to stabilizing holds of the same duration and effort level (Fig. 3, rightmost distributions) for both dorsiflexion (sombrero – hold: 1.24 pps [95% CI: 0.99–1.48]; $d = 1.02$) and plantarflexion (sombrero – hold: 1.34 pps [95% CI: 1.04–1.61]; $d = 1.06$). Additionally, the second plateau of the sombrero contraction exhibits significantly greater increases in dorsiflexion (sombrero – hold: 2.59 [95% CI: 1.45–3.72], $d = 1.34$) and

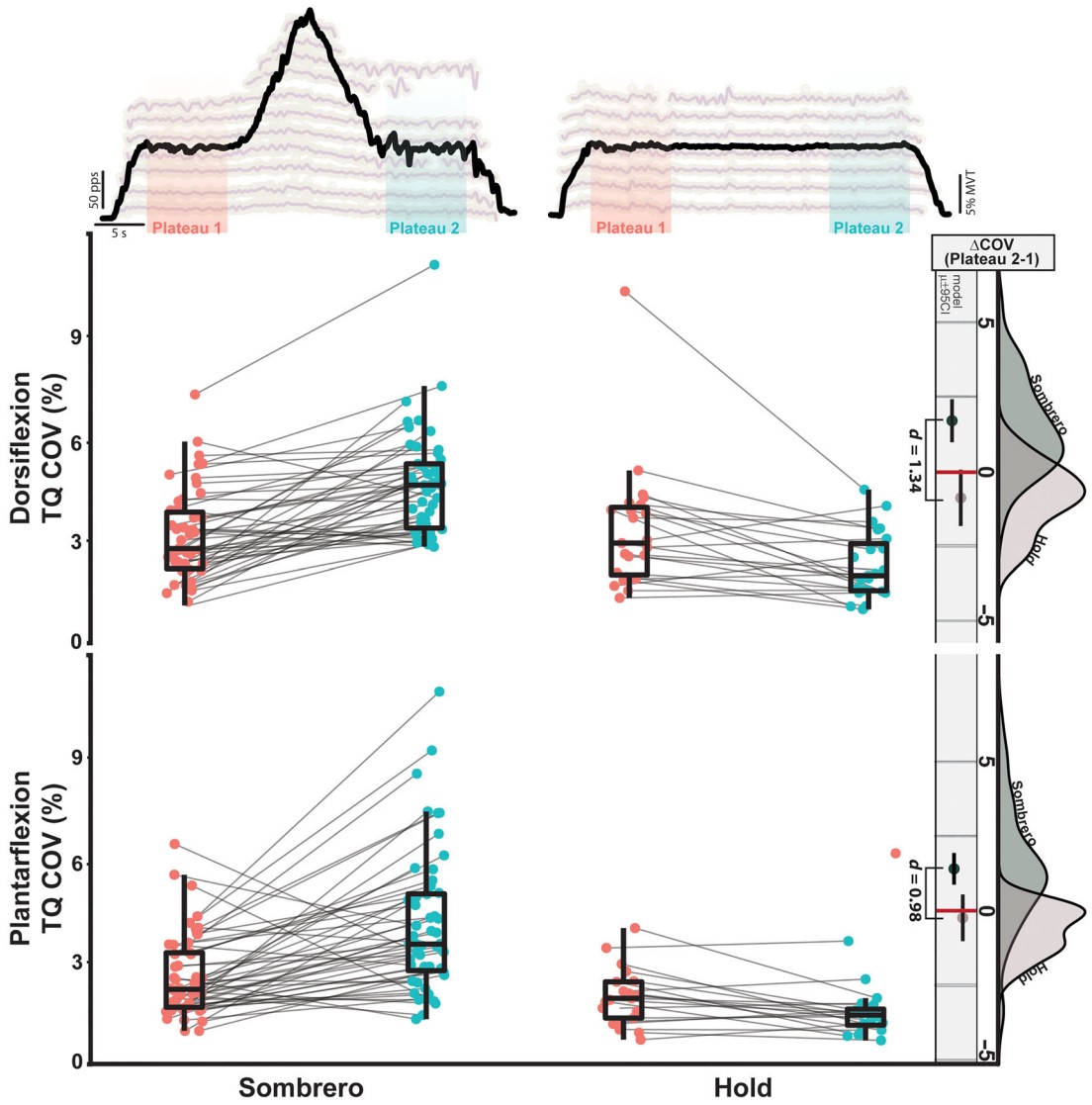

**Figure 2. Torque variability increases during the second plateau in the sombrero contraction**
Shown across the top are exemplar sombrero and hold contractions, with indicators for the first and second plateau region (plateau 1 is red, plateau 2 is blue). The centre panel represents the coefficient of variation (CV) in ankle torque for each trial in the first and second plateau. Lines connect trials. The rightmost column represents the change in CV between plateaus (CV at plateau 2 − CV at plateau 1). The rightmost distributions indicate the probability density of these values for both contractions. The vertical line and coloured circles indicate the model estimated mean and 95% confidence interval. Cohen's effect size *d* is shown between the two contractions when there is a significant difference. [Colour figure can be viewed at wileyonlinelibrary.com]

plantarflexion (sombrero – hold: 1.64 [95% CI: 0.69–2.59], $d = 0.98$) torque variability than the stabilizing holds (Fig. 2, rightmost distributions). Taken together, the sombrero contraction induces increased torque variability associated with more active MUs firing at lower average rates.

## Experiment 2: muscle length (joint angle) modulates PIC prolongation

To control the balance of excitatory and inhibitory input to motoneurons, we modulated the length of agonist–antagonist muscle pairs by changing the angle of the ankle (70°, 90° and 110°) prior to linear isometric dorsiflexion and plantarflexion ramp contractions. To account for alterations in biomechanical properties (e.g. moment arm) and cross-bridge dynamics of the agonist muscle we employed relative efforts, normalized to an individual's maximal abilities for a given

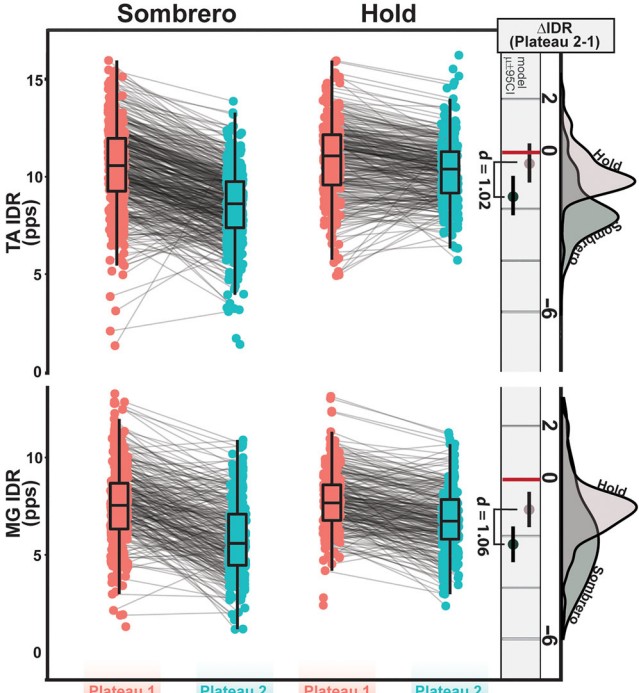

**Figure 3. Average MU discharge rate decreases during the second plateau in the sombrero contraction for only brim units** The centre panel represents the average MU discharge rate for each trial in the first and second plateau region. Lines connect independent MUs. The rightmost column represents the change in discharge rate between plateaus (IDR at plateau 2 — IDR at plateau 1). The rightmost distributions indicate the probability density of these values for both contractions. The vertical line and coloured circles indicate the model estimated mean and 95% confidence interval. Cohen's effect size *d* between the two contractions is shown when there is a significant difference (MG, medial gastrocnemius; TA, tibialis anterior; IDR, instantaneous discharge rate). [Colour figure can be viewed at wileyonlinelibrary.com]

ankle configuration. This yielded lower maximal torque generation for shorter muscles ($F = 24.09$, $P < 0.001$). That is, lower for dorsiflexion as the dorsiflexors were shortened (70°) and lower maximal plantarflexion torque as the plantar flexors were shortened (110°). These trends and quantitative values are shown in Table 1, with all isometric torque ramps normalized such that each was 30% MVT relative to the ankle configuration. The total number of decomposed MUs for each ankle configuration and muscle is given in Table 1.

Figure 4*B* shows all decomposed MUs for the MG and TA at each of the three ankle angles, highlighting a potentiation of PIC prolongation at short muscle lengths. For ease of interpretation, the angles have been categorized based on the length of the agonist muscle (e.g. 70° – long MG, short TA). For each respective plot, the smoothed discharge rates of individual MUs are offset in the *y*-axis and coloured according to the torque at which they were recruited, with the black trace indicating the average torque performed across all trials and participants. Greater prolongation of MU discharge is seen when a muscle is short, indicated with a greater number of teal-coloured higher threshold MUs that sustain discharge until the end of the ramp contraction. This indicates a shift in excitability and reduced ability to terminate PICs.

To quantify prolongation of MU discharge from PICs, we employed a paired MU analysis method ($\Delta F$; Fig. 4*A*). Across both muscles, for MUs matched between at least two lengths, we found muscle length to predict changes in $\Delta F$ ($\chi^2(4) = 33.24$, $P < 0.001$), with an estimated increase of 1.08 pps (95% CI: [0.47–1.69]) from long to short muscle lengths. Separating by muscle, from the longest to the shortest length, we estimate $\Delta F$ to increase by 1.29 pps (95% CI: [0.24–2.35]; $d = 0.99$) in the MG and 0.86 pps (95% CI: [0.24–1.48]; $d = 0.66$) in the TA (Fig. 4*C*). To characterize prolongation from PICs and their contribution to torque generation, we quantified the duration of MU discharge on the descending phase of each ramp and the torque that each MU was derecruited at (Fig. 4*A*). Across both muscles, we found muscle length to predict this descending duration ($\chi^2(4) = 69.58$, $P < 0.001$) and increase by 1.45 s (95% CI: [1.03–1.88]) from long to short muscle lengths. Separating by muscle, from the longest to the shortest length, we estimate descending duration to increase by 2.19 s (95% CI: [1.46–2.91]; $d = 1.82$) in the MG and 0.73 s (95% CI: [0.29–1.16]; $d = 0.61$) in the TA (Fig. 4*C*). Furthermore, the average derecruitment torque (TQ) of these MUs was also decreased. We found muscle length to predict the average torque at derecruitment ($\chi^2(4) = 64.94$, $P < 0.001$), and decrease from the longest to the shortest length by 5.04% MVT (95% CI: [3.28–6.79]; $d = 1.74$) in the MG and 1.80% MVT (95% CI: [0.75–2.86]; $d = 0.62$) in the TA (Fig. 4*C*).

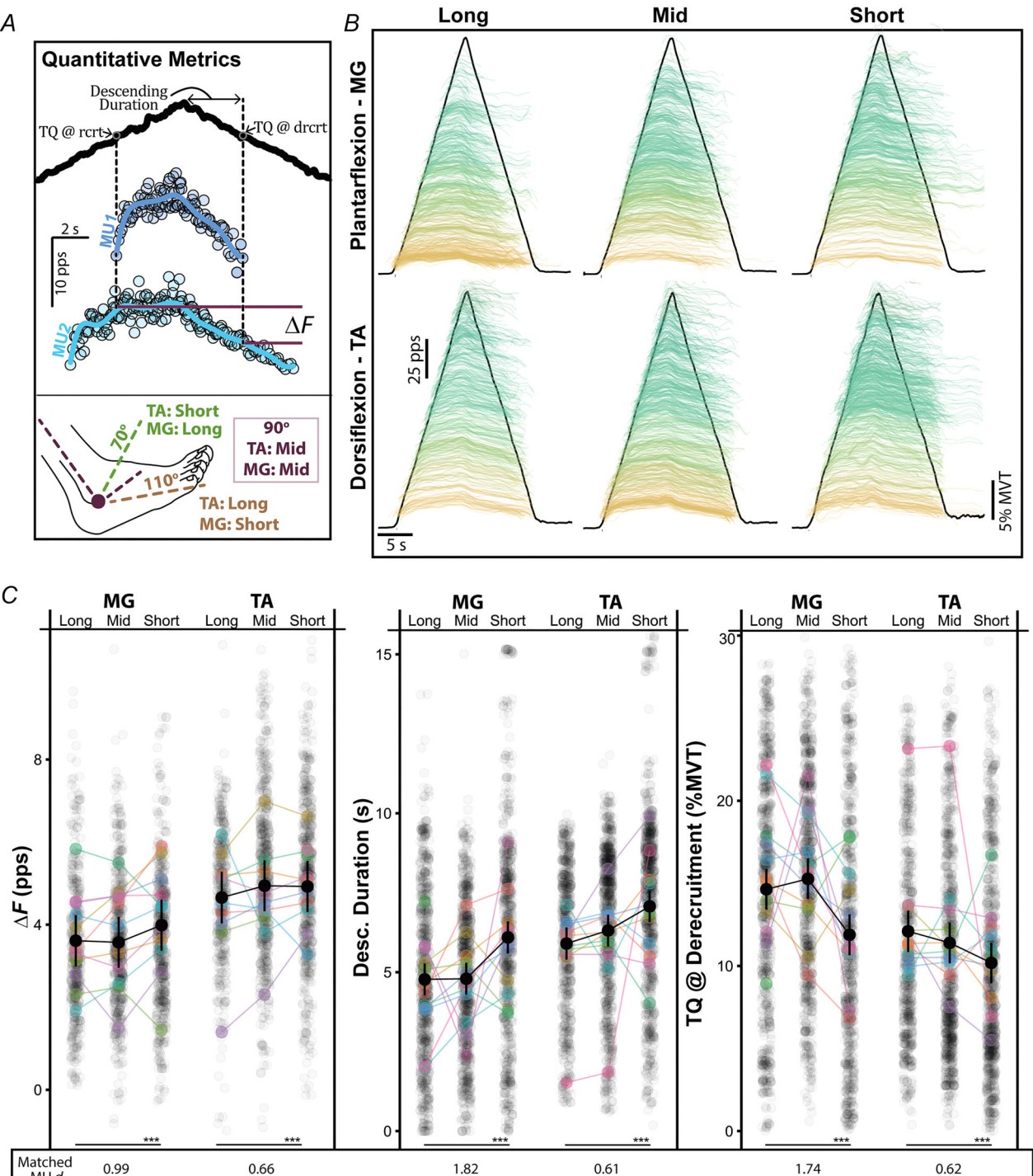

**Figure 4. Prolongation of discharge from PICs is greatest at short agonist muscle lengths**
*A*, the primary measures of PIC-induced prolongation are shown quantified for 'MU1'. The employed ankle angles, and corresponding muscle lengths for the tibialis anterior (TA) and medial gastrocnemius (MG), are shown in the bottom of *A*. In *B*, all decomposed motor units (MUs) are shown for the TA and MG at each muscle length. MUs are smoothed and both coloured and offset in the *y*-axis based on when they were recruited in the triangular contraction. The average dorsiflexion or plantarflexion torque (TQ) across all trials is indicated by the black trace. The quantitative metrics of PIC prolongation are shown for the entire population of decomposed MUs at each length in *C*. This includes the paired MU analysis (ΔF), the duration of time spent on the descending (desc.) portion of the ramp, and the TQ at which an MU is derecruited. Additionally, the effect size (Cohen's *d*) between the differences observed at the long and short lengths is shown across the bottom for matched MUs decomposed at a minimum of two lengths. [Colour figure can be viewed at wileyonlinelibrary.com]

**Table 1. Maximum voluntary torque (MVT) and number of decomposed motor units (MUs) as a function of muscle length. Matched MUs indicate units that were decomposed at a minimum of two independent lengths. Values indicate mean and SD.**

|  | Long | Mid | Short |
|---|---|---|---|
| **Maximum torque (Nm)** |  |  |  |
| Dorsiflexion | 24.38 (6.13) | 23.27 (6.31) | 15.62 (3.48) |
| Plantarflexion | 68.54 (26.63) | 57.38 (17.34) | 44.00 (11.24) |
| **MUs decomposed per trial** |  |  |  |
| TA | 12.74 (8.26) | 14.96 (8.86) | 17.07(8.48) |
| MG | 13.35 (7.37) | 13.17 (7.76) | 14.48 (9.59) |
| **Total matched MUs (no.)** | *Matched to mid or short* | *Matched to long or short* | *Matched to mid or long* |
| TA | 153 | 222 | 93 |
| MG | 66 | 77 | 37 |

Of note, although the data depicted in Fig. 4*B* and *C* include the entire dataset, the primary statistics that we report are based on MUs matched at a minimum of two independent lengths (see Fig. S1). The trends displayed in Fig. 4 remain with the matched MUs and all metrics for the matched MUs can be observed in the supplementary information.

## Experiment 3: greater PIC prolongation and higher torque variation at short muscle lengths

Given the degradation of torque control observed alongside greater PIC prolongation in experiment 1 and the modulation of PIC prolongation with muscle length in experiment 2, we sought to show that changes in muscle length could manipulate the observed degradation in torque control. If PIC prolongation does indeed generate the amplified torque variability observed in experiment 1, facilitating this behaviour through shortening an agonist muscle should further degrade torque control, and vice versa. To investigate this relationship, we had individuals perform both the superimposition sombrero task and long stabilizing hold contractions of experiment 1 (see Fig. 2) at the two extreme muscle lengths tested in experiment 2. When short, we decomposed 1123 MUs for MG and 1051 MUs for TA while when long we decomposed 1005 MUs for MG and 908 MUs for TA.

An increase in PIC prolongation behaviour can be seen in Fig. 5*A* at shorter muscle lengths, with brim, cap and button MUs coloured as indicated. A greater number of cap MUs is observed at shorter lengths, with their continuous discharge estimates shown in red and often lasting through the second plateau. This increase in cap MUs at short muscle lengths can additionally be appreciated in Fig. 5*B* with the increase in red data points, and in Fig. 5*D* with the average number of cap MUs per trial. Across both muscles, we found muscle length to significantly predict the number of cap MUs ($\chi^2(4) = 55.54$, $P < 0.001$), with an estimated 5.17 (95% CI: [2.60–7.75]) more cap MUs per trial for the MG and 4.41 (95% CI: [1.75–7.10]) per trial for the TA (Fig. 5*D*) at shorter lengths. Additionally, the duration of sustained discharge for these cap MUs can be observed in Fig. 5*B*, with the proportion of units that sustain discharge for more than 2 s into the second plateau significantly greater at shorter lengths ($\chi^2(2) = 49.09$, $P < 0.001$) for both dorsiflexion ($d = 0.98$) and plantarflexion ($d = 1.38$).

Alongside greater cap MUs at shorter muscle lengths, task performance in the second plateau worsens at shortened muscle lengths. When comparing the increase in CV in ankle torque from *plateau one* to *plateau two* (Fig. 5*C*; $\Delta$TQ CV), we found muscle length to predict this increase in variability ($\chi^2(1) = 6.169$, $P = 0.013$, $d = 0.39$) across muscles. Separating by muscle, we found shorter muscle lengths to produce the highest increases in CV, with estimated increases of 2.00% (95% CI: [1.24–2.76]) for dorsiflexion and 1.70% (95% CI: [0.98–2.42]) for plantarflexion. In comparison, at the longer muscle lengths, CV was increased by an estimated 1.13% (95% CI: [0.37–1.89]) for dorsiflexion and 0.84% (95% CI: [0.10–1.58]) for plantarflexion.

Interestingly, though the general trend of lower average discharge rates in plateau two remained, this change was greatest at long muscle lengths. Specifically, despite a greater number of cap MUs actively discharging in plateau two at shorter muscle lengths, we found brim MU discharge rates to decrease by less (Fig. 5*C*) and cap MU discharge rates to increase (Fig. 5*D*). Across muscles, we found muscle length to predict the change in brim MU discharge rate ($\chi^2(4) = 521.12$, $P < 0.001$), with shortest lengths producing the lowest decrease between plateaus (Long – MG: 2.78 pps [95% CI: 2.28–3.27], TA: 2.40 pps [95% CI: 1.92–2.88]; short – MG: 0.803 pps [95% CI: −0.32 to 1.29], TA: 0.70 pps [95% CI: 0.22–1.18]). Furthermore, we found muscle length to predict cap MU discharge rates ($\chi^2(2) = 23.16$, $P < 0.001$), increasing by 1.26 pps (95% CI: [0.69–1.84]) across the MG and TA at shorter muscle lengths.

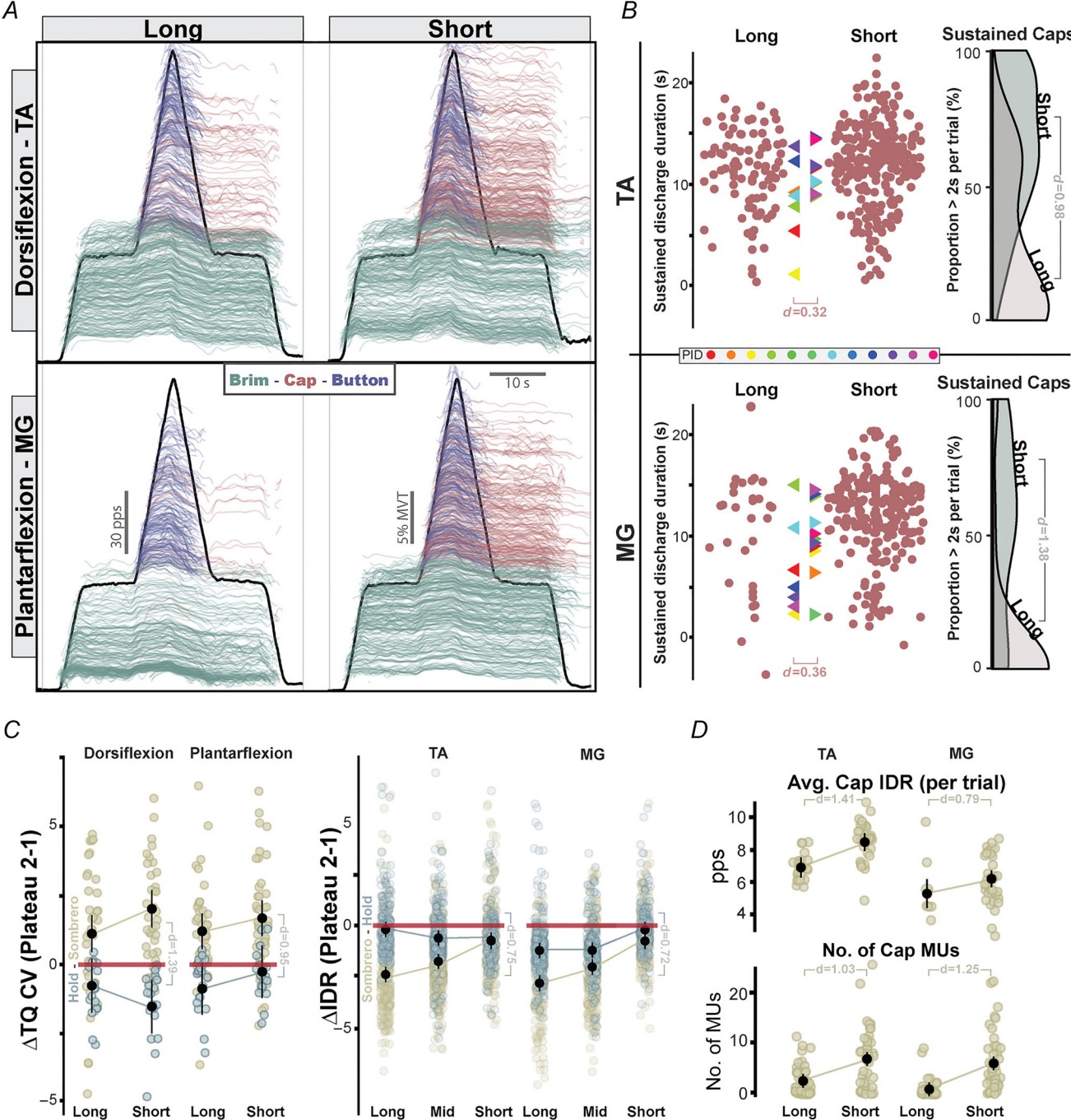

**Figure 5. Greater PICs and torque variability at shorter muscle lengths in the sombrero contraction**
*A*, the entire population of decomposed motor units (MUs) for the tibialis anterior (TA) and medial gastrocnemius (MG) at long and short muscle lengths. MUs are smoothed and offset in the *y*-axis based on when they were recruited in the contraction. The average dorsiflexion or plantarflexion torque across all trials is indicated by the black trace. MU categories are indicated by colours. In the left portion of *B*, the duration of sustained discharge past a theoretical derecruitment is shown for all cap MUs, with coloured triangles indicating participant averages. In the rightmost portion of *B*, probability density distributions for the proportion of MUs that turn on in the centre ramp that sustain discharge 2 s into the second plateau are shown for both lengths. The change in CV (plateau 2 − plateau 1) is shown in the left panel of *C* for the sombrero and hold contractions. In the right panel of *C*, the average change in brim MU discharge rate (plateau 2 − plateau 1) is shown. The black vertical lines and dots represent the model estimated means and 95% confidence interval. In *D*, the average cap MU discharge rate for each trial is shown in the top row and the number of cap MUs for each trial is shown in the bottom row. In all, Cohen's *d* values and pointers indicate a comparison with significant differences. [Colour figure can be viewed at wileyonlinelibrary.com]

Comparing stabilizing hold and sombrero contractions, we found sombreros to exhibit significantly greater variation in ankle torque ($\chi^2(4) = 66.98$, $P < 0.001$) for both dorsiflexion (sombrero-hold: Long 1.93% [95% CI: 0.46–3.40]; Short 3.59% [95% CI: 2.12–5.06], $d = 1.39$) and plantarflexion (sombrero-hold: Long 1.79% [95% CI: 0.38–3.19]; Short 1.97% [95% CI: 0.55–3.39], $d = 0.95$), but no interaction between contraction type and length ($\chi^2(3) = 7.12$, $P = 0.07$). Furthermore, we found contraction type to predict the change in brim MU discharge rate between plateaus ($\chi^2(6) = 431.22$, $P < 0.001$), being lower during the sombrero for all but the shortest muscle length for dorsiflexion (sombrero-hold: Long −2.21 [95% CI: −2.49 to −1.94]; Mid −1.11 [95% CI: −1.35 to −0.87]; Short −0.06 [95% CI: −0.23 to 0.35], $d = 0.75$) and plantarflexion (sombrero-hold: Long −1.60 [95% CI: −1.87 to −1.32]; Mid −0.86 [95% CI: −1.13 to −0.59]; Short −0.69 [95% CI: −0.96 to −0.41], $d = 0.72$).

## Discussion

Dendritic PICs in motoneurons introduce a monoaminergic-dependent amplification and prolongation of excitatory synaptic inputs that require task-dependent control, so that they do not engender aberrant discharge patterns and poorly controlled forces. In this study, we demonstrate through a series of experiments that isometric paradigms that conceivably restrict the ability of inhibitory inputs to control PICs exacerbate prolongation of motoneuron discharge and introduce subsequent problems in torque control.

### Prolongation of discharge from PICs degrades torque control

In the first paradigm, we asked individuals to perform a superimposition task (i.e. sombrero contraction) that challenged the inhibitory input necessary to deactivate PICs. In short, this superimposition task is composed of an isometric ramp atop a low-effort stabilizing hold, where individuals were to perform the ramp contraction and return to the stabilizing hold as precisely as possible (see Fig. 1A). Upon returning to the hold (plateau two), the inhibitory input necessary to deactivate the PICs of MUs recruited during the ramp contraction was compromised as continued excitatory drive is necessary to accomplish the stabilizing hold. Subsequently, many of these newly recruited MUs sustained discharge (i.e. PIC prolongation) and became what is termed cap MUs (Fig. 1A). This created a situation where higher threshold MUs were actively discharging in the second plateau, which forced the entire motor pool to discharge at a lower average rate (Fig. 3) and increased torque CV (Fig. 2).

The notion that cap MUs, and thus prolongation from PICs, are responsible for higher torque variation in plateau two of the sombrero contraction is supported by findings from both the first and third experiments. In both experiments, we found that stabilizing holds of the same duration did not exhibit cap MUs nor an increase in torque CV. While the hold contractions elicited decreased torque CV (i.e. improved performance) for dorsiflexion and no change during plantarflexion, CV was increased by 1.5-fold in the sombrero contractions. Furthermore, in the third experiment, we found muscle length-induced increases in cap MUs and PIC prolongation to occur alongside increases in torque variability.

Although experiments 1 and 3 suggest that cap MUs introduce difficulties in torque control, a precise mechanism remains elusive. Potentially, the presence of cap MUs could put motor output in a higher gain state, where a similar descending excitatory drive produces larger changes in MU discharge and entrains many more units (see Supplementary experiment 1). Indeed, prior work has shown that increased motoneuron gain amplifies noise in motor commands (Wei et al., 2014). To investigate this interpretation, we completed a series of simulations using realistic motoneuron models to probe excitability on the first and second plateau of the sombrero. These simulations showed that identical excitatory synaptic input produced net output in motoneuron discharge twice as large in the second plateau. Thus, in addition to amplifying synaptic noise, these larger deviations in discharge rate may lead to a scenario where individuals tend to overshoot/overcorrect and create larger oscillations in torque about the target level. Regardless, this increase in variability is quite substantial, has recently been shown to occur across a range of force output levels and necessitates further exploration (Darendeli & Enoka, 2024). This increase in variability probably represents a meaningful degradation in control, as force steadiness and CV are thought to be a proxy for neural drive to the motor pool and predictive of motor function in health and disease (Enoka & Farina, 2021).

### Muscle length (joint angle) modulates prolongation from PICs

Altering ankle joint angle, and subsequently muscle length, has long been appreciated to modulate measures of spinal reflex excitability, with shorter ankle muscles producing greater electrophysiological estimates of spinal excitability (Burke et al., 1983; Gerilovsky et al., 1977, 1989). In addition to greater $M_{max}$ and $H_{max}$ at shorter muscle lengths, both plantar and dorsiflexor muscles exhibit greater $H_{max}/M_{max}$ ratios, indicating greater spinal reflex pathway excitability irrespective of peripheral muscle mechanisms (Dutt-Mazumder

et al., 2020; Frigon et al., 2007; Hwang, 2002; Patikas et al., 2004). Explanations and discussions regarding these observations are varied and have included altered motoneuron excitability, presynaptic inhibition of Ia afferents, muscle fibre volume, Golgi tendon organ (Ib) feedback and the relation of muscle fibres to the skin (Frigon et al., 2007; Garland et al., 1994; Gerilovsky et al., 1989; Hwang, 2002; Patikas et al., 2004).

We speculate that the observed changes in prolongation across muscle length in the second experiment are due to changes in inhibitory input to motoneurons. This speculation is based upon our observations that MUs have longer average durations of discharge on the descending limb of the ramp and lower average derecruitment torques at shorter muscle lengths for both the entire decomposed MU population (Fig. 4C) and for MUs identified during at least two length conditions (i.e. matched MUs; Fig. S1). Analysing the discharge trajectories of matched MUs indicates a rightward shift in MU activity, with later recruitment and derecruitment and increased time on the descending phase.

PICs are particularly sensitive to inhibitory inputs, in part due to their dendritic location, being deactivated by both recurrent and reciprocal inhibition (Bui et al., 2008; Hyngstrom et al., 2007; Kuo et al., 2003). Importantly, the control of PICs is probably only achieved via inhibitory inputs, as excitatory inputs (e.g. Ia muscle spindle afferents) only activate PICs, which remain active even after cessation of this excitatory input. It is possible that changes in ankle angle could alter inhibitory input via multiple means. In addition to altering agonist muscle length and discharge of spindle afferents, changes in ankle angle also alter antagonist muscle length (i.e. reciprocal inhibition), passive tension on the tendon, joint and cutaneous feedback, and heteronymous input from synergist muscles, all of which could alter synaptic input to agonist motoneurons (Baxendale & Ferrell, 1982; Crone et al., 1987; Fallon et al., 2005; Hongo et al., 1984; Hunt, 1952; Nichols, 2018; Pierrot-Deseilligny & Burke, 2012).

Furthermore, since we employed relative efforts and the absolute forces required for the ramp contractions at each muscle length were necessarily different, Golgi (Ib) feedback may be altered and could explain the observed results. Though Ib feedback is complex and less understood than Ia feedback (Pierrot-Deseilligny & Burke, 2012), given that the ramp contractions at shorter muscle lengths were performed at absolute values of torque lower in magnitude, Ib inhibition could be lower (Houk et al., 1971; Jami, 1992). This would lower the overall inhibitory drive to motoneurons and allow for greater prolongation from PICs. Similarly, greater Ib inhibition at longer muscle lengths when both the absolute force generation for the contraction and the passive tendon/muscle tension is greater could reduce the effective drive of the dendritic PICs.

## Further degradations in torque control with greater prolongation from PICs at short muscle lengths

In the third and final experiment of this study, we combined the superimposition task of the first experiment and the extreme changes in muscle length of the second experiment. Combining these paradigms, we again observed the prolonging behaviour of PICs to be greatest at short muscle lengths (Fig. 5A, B, D) and found this behaviour alongside increases in torque CV (Fig. 5C).

At short muscle lengths we found the propensity for sustained discharge of MUs recruited in the ramp portion of the sombrero to be markedly increased (see Fig. 5A, B). When observing the proportion of these newly recruited MUs that sustained discharge more than 2 s past their theoretically expected derecruitment, a large increase in the average proportion of MUs is observed at short lengths ($d = 0.98$, 1.38 for TA, MG). This simultaneously strengthens the observations of greater prolonging behaviour at shorter lengths found in the second experiment while amplifying the negative ramifications of this behaviour in the sombrero from the first experiment.

We reason that the greater number of cap MUs actively discharging in the second plateau at short muscle lengths is responsible for the increased torque CV observed in the sombrero contraction. To perform the centre ramp portion of the sombrero, higher threshold MUs are recruited which then subsequently sustain discharge into the second plateau (i.e. cap MUs). As descending excitatory drive to the motor pool is progressively decreased to perform the descending portion of the ramp, the sustained discharge in these cap MUs is probably maintained primarily by the intrinsic current from PICs. Since PICs are keeping cap MUs above their activation threshold, descending drive may similarly entrain these MUs and thus their discharge rates could represent oscillations in common input to the motor pool. We speculate that this oscillation may introduce difficulties in control as variations in descending drive now entrain many more MUs, particularly a greater proportion of higher threshold MUs with relatively larger muscle twitches and greater force production capacity (Henneman & Mendell, 2011). Specifically, oscillations in the discharge rate of these higher threshold cap MUs generate greater deviations in torque that could lead to complexities in control (e.g. over-corrections). This speculation is further corroborated by the results of supplementary experiment 1, where the deviation in discharge rate is over twice as large for excitatory synaptic inputs during the second plateau.

As was speculated and observed in the first experiment, the average discharge rate of the entire motor pool decreased during the second plateau of the sombrero contraction, likely to accommodate the greater number

of actively discharging MUs. Interestingly, though brim units still generally exhibited a decrease in discharge rate from plateau one to two (negative values, Fig. 5*C*), these values became progressively less negative at shorter muscle lengths. Similarly, the average discharge rate of cap MUs displayed a relative increase at short lengths, despite significantly greater cap MUs actively discharging (Fig. 5*D*). Though seemingly peculiar, as now many more MUs are actively discharging at lower rates to produce the same relative (lower absolute) force, this is probably due to a complex interaction with changing MU twitch properties. It has been shown that the twitch course of MUs possess a shorter duration and half-relaxation time constant when a muscle is shortened. Thus, at shorter lengths, lower discharge rates may produce less effective summation of each MU's twitches, reducing the smoothness of their force outputs and contributing to the overall increase in torque CV (Bigland-Ritchie et al., 1992; Rack & Westbury, 1969; Vander Linden et al., 1991).

## The sombrero contraction as an estimate of prolongation from PICs

While we used the sombrero contraction primarily to emphasize the critical role of inhibitory control in regulating PICs, this contraction also serves as a valuable quantitative tool for estimating PIC-induced prolongation. Traditionally, PICs in human motor units are assessed using ramp contractions, as demonstrated in experiment 2, which was specifically designed to examine how muscle length influences PIC prolongation. However, the sombrero contraction provides additional insights by highlighting the propensity for sustained discharge from PICs and its functional implications for torque production. Unlike ramp contractions, the sombrero allows for direct quantification of how many motor units remain active due to PICs and the duration of their sustained discharge.

## Implications: neurological impairment and muscle cramps

The findings presented here suggest that difficulties in human torque control arise when constraints are placed on the inhibitory control of PICs. Though long appreciated for their critical role in motor control, PICs may introduce problems in motor control at the level of the motoneuron when improperly managed. This is particularly relevant to neurological conditions where pathological shifts in monoamines have been theorized (e.g. spinal cord injury, chronic stroke). Monoamines amplify PICs by as much as five-fold, markedly potentiating PIC amplification and prolongation of discharge. Thus, when monoaminergic drive to the cord is pathologically high, such as is theorized

in chronic hemiparetic stroke (Beauchamp et al., 2024; McPherson, McPherson et al., 2018), large PICs could make motoneurons hyperexcitable. These hyperexcitable motoneurons, being more sensitive to afferent inputs, may lead to spasticity and even hypertonicity, where individuals are unable to derecruit and cease motoneuron discharge. Furthermore, more excitable motoneurons also amplify the commands of weak and indirect motor pathways (e.g. reticulospinal) which is theorized to generate a loss of independent joint control (Beauchamp, Hassan et al., 2023; Li et al., 2019; McPherson & Dewald, 2022).

Beyond neurological conditions, findings from the last two experiments may speculatively suggest a link between PICs and muscle cramps. Muscle cramps, particularly in the calf (triceps surae), are more common in shortened muscles and can be relieved by transient inhibitory inputs (Khan & Burne, 2007; Mills et al., 1982). Given the sensitivity of PICs to inhibitory inputs, and their ability to generate prolonged MU discharge, they may contribute to cramp generation. While this idea has been proposed (Baldissera et al., 1991; Khan & Burne, 2007; Minetto et al., 2013), direct evidence has been lacking. This study provides an initial link, showing that PIC-related prolongation is most pronounced at shorter muscle lengths (experiment 2) and may impair voluntary MU derecruitment (experiment 3), potentially leading to sustained contraction.

## Considerations and limitations

Several considerations and limitations must be appreciated when interpreting this work. First, the findings are necessarily limited by the population of MUs decomposed with HDsEMG and may not represent the entire motor pool. The decomposition of surface EMG is biased towards larger MUs and MUs closer to the electrode (Caillet et al., 2023; Farina & Holobar, 2016). Additionally, the participant population was predominantly female, preventing the assessment of potential sex differences (Jenz et al., 2023). Futhermore, as noted, changes in joint angle result in many changes in addition to changes in muscle length (i.e. afferent input, descending drive). Although antagonist EMG was monitored throughout the contractions and no substantial deviations were observed, a contribution from the antagonist to the observed results cannot be ruled out. Lastly, the matching of MU action potentials in experiment 2 is not without potential error. Importantly, changes in the action potential shapes as a function of muscle length could certainly introduce errors. That said, the matched data in the supplementary materials and unmatched data shown in Fig. 4*C* indicate the same trends and findings.

Notably, this work fails to replicate similar work in the decerebrate cat. Studies from our group in the decerebrate

cat show that PICs in agonist motoneurons are markedly potentiated as a result of reduced reciprocal inhibition when its antagonist muscle is shortened (Hyngstrom et al., 2007). Though this is in direct contrast to the findings of this study, the decerebrate cat lacks descending corticospinal input and contractions in the prior study were not volitional. Of note, reciprocal Ia inhibitory postsynaptic potentials in the baboon were found to be evoked only after spinal transection or depressed brain function, implying a descending tonic inhibition of reciprocal Ia afferents (Hongo et al., 1984). Perhaps humans exhibit similar descending inhibitory control, which could explain the divergence of our results from the decerebrate cat and potentially implicate changes in descending presynaptic Ia inhibition as a contributing mechanism. Indeed, a reduction in pre-synaptic Ia inhibition at shortened muscle lengths may be advantageous given the deprecated force production capacity of shortened muscles.

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

## Additional information

### Data availability statement

All data are available upon reasonable request.

## Competing interests

None declared.

## Author contributions

All data were collected at Northwestern University. Conceptualization: J.A.B., G.E.P., O.U.K., F.N., J.P.A.D., C.J.H.; Methodology: J.A.B., G.E.P., O.U.K., F.N., J.P.A.D., C.J.H.; Investigation: J.A.B., G.E.P., O.U.K.; Visualization: J.A.B.; Writing – original draft: J.A.B.; Writing – review & editing: J.A.B., G.E.P., O.U.K., F.N., J.P.A.D., C.J.H.

## Funding

NIH/NINDS F31NS120500 to J.A.B., NIH/NICHD R01HD039343 and NIH/NINDS R01NS105759 to J.P.A.D., NIH/NINDS R01NS125863 to C.J.H.

## Keywords

impairment, motor control, motoneurons, motor unit, persistent inward current, sombrero

## Supporting information

Additional supporting information can be found online in the Supporting Information section at the end of the HTML view of the article. Supporting information files available:

**Peer Review History**
**Supplementary Information**

