## [Peer Review History · The Journal of Physiology]

Intrinsic properties of spinal motoneurons degrade ankle torque control in humans

James A Beauchamp, Gregory EP Pearcey, Obaid U Khurram, Francesco Negro, Julius P.A. Dewald, and C. J. Heckman
DOI: 10.1113/JP287446

Corresponding author(s): C. J. Heckman (c-heckman@northwestern.edu)

Review Timeline:

Submission Date:	05-Aug-2024
Editorial Decision:	01-Oct-2024
Revision Received:	10-Feb-2025
Accepted:	26-Feb-2025

Senior Editor: Richard Carson

Reviewing Editor: Madeleine Lowery

Transaction Report:

Dear Dr Heckman,

Re: JP-RP-2024-287446 "Intrinsic properties of spinal motoneurons degrade ankle torque control in humans" by James A Beauchamp, Gregory EP Pearcey, Obaid U Khurram, Francesco Negro, Julius P.A. Dewald, and C. J. Heckman

Thank you for submitting your manuscript to The Journal of Physiology. It has been assessed by a Reviewing Editor and by 2 expert referees and we are pleased to tell you that it is potentially acceptable for publication following satisfactory major revision.

REVISION CHECKLIST:

Please upload two versions of your manuscript text: one with all relevant changes highlighted and one clean version with no

changes tracked. The manuscript file should include all tables and figure legends, but each figure/graph should be uploaded as separate, high-resolution files.

We look forward to receiving your revised submission.

Yours sincerely,

Richard Carson
Senior Editor
The Journal of Physiology

REQUIRED ITEMS

- Author photo and profile. First or joint first authors are asked to provide a short biography (no more than 100 words for one author or 150 words in total for joint first authors) and a portrait photograph. These should be uploaded and clearly labelled together in a Word document with the revised version of the manuscript. See Information for Authors for further details.
- You must start the Methods section with a paragraph headed Ethical Approval. If experiments were conducted on humans, confirmation that informed consent was obtained, preferably in writing, that the studies conformed to the standards set by the latest revision of the Declaration of Helsinki and that the procedures were approved by a properly constituted ethics committee, which should be named, must be included in the article file. If the research study was registered (clause 35 of the Declaration of Helsinki), the registration database should be indicated, otherwise the lack of registration should be noted as an exception (e.g. The study conformed to the standards set by the Declaration of Helsinki, except for registration in a database). For further information see: <https://physoc.onlinelibrary.wiley.com/hub/human-experiments>.
- Your manuscript must include a complete Additional Information section, including competing interests; funding; author contributions and acknowledgements.
- Please upload separate high-quality figure files via the submission form.
- Please ensure that the Article File you upload is a Word file.
- Papers must comply with the Statistics Policy: https://jp.msubmit.net/cgi-bin/main.plex?form_type=display_requirements#statistics.

In summary:

- If $n \leq 30$, all data points must be plotted in the figure in a way that reveals their range and distribution. A bar graph with data points overlaid, a box and whisker plot or a violin plot (preferably with data points included) are

acceptable formats.

- If $n > 30$, then the entire raw dataset must be made available either as supporting information, or hosted on a not-for-profit repository, e.g. FigShare, with access details provided in the manuscript.
- 'n' clearly defined (e.g. x cells from y slices in z animals) in the Methods. Authors should be mindful of pseudoreplication.
- All relevant 'n' values must be clearly stated in the main text, figures and tables.
- The most appropriate summary statistic (e.g. mean or median and standard deviation) must be used. Standard Error of the Mean (SEM) alone is not permitted.
- Exact p values must be stated. Authors must not use 'greater than' or 'less than'. Exact p values must be stated to three significant figures even when 'no statistical significance' is claimed.

- Please include an Abstract Figure file, as well as the Figure Legend text within the main article file. The Abstract Figure is a piece of artwork designed to give readers an immediate understanding of the research and should summarise the main conclusions. If possible, the image should be easily 'readable' from left to right or top to bottom. It should show the physiological relevance of the manuscript so readers can assess the importance and content of its findings. Abstract Figures should not merely recapitulate other figures in the manuscript. Please try to keep the diagram as simple as possible and without superfluous information that may distract from the main conclusion(s). Abstract Figures must be provided by authors no later than the revised manuscript stage and should be uploaded as a separate file during online submission labelled as File Type 'Abstract Figure'. Please also ensure that you include the figure legend in the main article file. All Abstract Figures should be created using BioRender. Authors should use The Journal's premium BioRender account to export high-resolution images. Details on how to use and access the premium account are included as part of this email.

Reviewing Editor's comments:

The reviewers are in agreement on the contribution of the manuscript and the significance of the results presented. However, both have indicated that a degree of rewriting and restructuring is required before the manuscript would be suitable for publication. This would help to clarify the main hypotheses and contribution in terms of new physiological insights as well as guiding the reader through the results. There are also a number of specific questions and comments to be addressed.

Referee #1:

This study examined a motor control paradox that may exist in persistent inward current function for motoneurons, where the known amplification of currents and prolongation of discharge may not be mutually beneficial for all motor control tasks. I read this manuscript with great interest, and I believe that the outcomes of the study will be of great interest to the wider

motoneurone and physiology community. The team that performed this work has an excellent track record in the field and they have produced another excellent series of experiments. They present a novel contraction task referred to as 'sombbrero' contractions, which members of this team were the first to present in preprint (BioRxiv 2023) and was employed by the Kavanagh lab (J Physiol, 2024) and the Enoka lab (Exp Brain Res, 2024).

My major comment lies in the style of writing, as I think there needs to be changes made to the manuscript which remove the colloquial language entirely, and greater emphasise the physiology underpinning the experiments. For example, the introduction provides a good overview of background literature, as well as areas that require further research, but as a reader I could not identify what the current experiment was testing. There was no purpose or hypotheses to guide the reader into the experiments that have been designed for the current study. Hence, the majority of my comments are based on clarity rather than quality of results (which are excellent). Similarly, there are several interpretations of physiological mechanisms in the results which would be better suited to the discussion. I believe with some further work on the clarity in the body text of the manuscript this would be an excellent contribution to physiology literature.

Abstract

It would be worth considering the balance of the abstract, as half of it is currently devoted to the background of PICs and brainstem neuromodulation, and only a sentence or two is devoted to the actual experiments and results.

Introduction

Line 79: Can there be more explanation provided for boosting currents. Not really clear from the text.

Line 86: Do monoamines adapt excitability? What is normal motor repertoire?

Line 88: not clear what this sentence is saying. What is tension?

Line 98: not clear what the perceptual error and conflicting inputs relate to. Is this statement necessary for the current study?

Please include a purpose and hypotheses so the reader can fully appreciate what experiment design and methods should be used. Some of these details are in the first few lines of the methods, but they could be even more explicitly related to the content of the introduction.

Methods

Line 129: Needs to state how many trials/attempts were used to obtain MVC, as well as the method (I'm assuming peak torque for all valid attempts).

The z-coherence bands do not align with most other studies use. Importantly, the authors have selected cut-offs at 10 Hz, whereas nearly all other studies have the band to quantify coherence associated with 10 Hz (e.g. 5-15 Hz). Physiological tremor arises from afferent activity and oscillatory activity at the spinal cord level, and manifests at 10 Hz for healthy individuals. Thus, the cut-offs won't help in understanding a key neural contributor to force characteristics.

Results

I strongly recommend removing the methods-based content from the results. The inclusion of methods not only doubles-up on the existing methods section, but also creates a very long results section to read. As one example, the very first paragraph of the results could have lines 347 to 356 deleted and the actual results would still remain.

There are several instances of interpretation in the results. I would recommend reporting statistically-driven results in this section, and move physiological interpretations to the discussion so that all interpretations can be viewed together.

I also recommend tightening up the language to avoid superlatives. For example, the terms stark and appreciated in line 498 "A stark increase in PIC prolongation behavior can be appreciated in Figure 5a". Many sentences like this can be edited throughout the manuscript. There are also many instances where 'degrade', or 'impact' or 'good control' are used which distract from the excellent graphs and reporting of data.

There is only coherence reported for the delta band and no mention of any other band? If this data has been analyzed as per the methods it is important to present it in results. I understand that delta coherence is the main determinant of force control during steady state contractions, but the other bands are still important for understanding how the CNS regulated

motoneurone discharge.

Despite these comments, I must commend the authors on the quality of their graphs and illustrations. They are superb.

Discussion

Once again I would recommend clear descriptions of findings. Statements such as 'we showcase', 'introduce impediments', 'engender aberrant discharge patterns', 'poorly controlled', 'challenging the inhibitory control' and 'problems in torque control' are not as informative as a direct description of the outcomes. These phrases are all in the first paragraph of the discussion. A few instances throughout the discussion are fine but not repeatedly.

Referee #2:

The overall finding reported in this manuscript is that the prolongation of motor unit discharges from the activation of PICs increases the coefficient of variation of force. This makes sense as larger motor units are still active at a lower force level than they would typically be recruited at, increasing the variability of the force. The secondary finding is that prolongation (and level of PICs), and thus CV of the force, is increased in conditions where there is likely to be less motor unit inhibition (shortened muscle lengths). This study shows that although PICs can aid in force production, they can also increase the variability of force produced. This has implications for conditions in which PICs are altered due to pathological shifts in monoamines.

These findings are easy to understand and explain in terms of the underlying physiology, but at the moment the main message of the paper is obscured by the number of results presented. Some restructuring of how the results are presented, and perhaps moving some supplementary analyses to the appendix/supplementary material, should help to distinguish the main message.

Main comments:

1. At the moment it is difficult to see how the three experiments fit together from the introduction and methods, and the description of the experiments is a bit disjointed. Their relationship is clearer after reading the results, but it would be better to bring this in earlier in the paper. Some restructuring would be helpful so that the combined aim is more obvious earlier in the manuscript, but more importantly so that it is easy to see how each experiment contributes to this aim. In the introduction or methods it could be stated whether multiple experiments are being used to test the same hypothesis, or whether each tests a unique aspect of the overarching hypothesis.

Although the introduction provides a nice overview of PICs, more it could be dedicated to a specific discussion of the background needed to support the aims of this study. Although there is a hypothesis given, the overarching aim (and specific aims of each sub-experiment) could be briefly mentioned here. The methods could also be restructured to make the aims a bit clearer. One suggestion is to include a very brief outline of the methods used for each experiment in the first paragraph (mirroring the aims that could be included at the end of the introduction). The more detailed information can then be addressed later in the methods. Some signposting indicating which sections will outline methodology shared between the experiments and which is specific to a particular experiment could also be helpful.

2. There are lots of different methods presented: sombrero profile, delta F, motor unit coherence, motor unit matching and model simulations. Dividing the focus of the paper over all of these different methodologies may be confusing the main message. Are all of these methodologies essential to the results of the paper and are they all needed in the main body of the paper? This may be the first time the sombrero contraction is presented (for peer reviewed research), so it would be good to make sure it is the focus.

3. Changing the muscle length would presumably alter the motor unit action potential waveform recorded at each electrode which would make matching based on action potential shapes very difficult to do accurately. This could be mentioned as a

limitation. In this case, I think visual inspection of matched MUs is also needed for the challenging scenario of matching across different muscle lengths. Some examples of matched MUAP templates in the supplementary material could also help this. Furthermore, it's not clear from the methods if it would be possible to have matches that include the same motor unit (or pair of motor units) multiple times or repeated pairs/motor units are excluded.

4. For the statistical analysis, if the number of motor units per trial differed significantly this could have a large effect on the results (as this number is being used to normalize the results). Another way of looking at this would be to use the number of motor units as a factor in the linear mixed model rather than using it to normalise the number of cap MUs. Does the result of a higher number of cap MUs at shorter muscle lengths still hold in this case? Also, is the delta F measure affected by the number of motor units decomposed? This is also something that could be investigated using the LMM.

Minor:

There are over 100 references, this number could be reduced. Some statements are supported by 6 references, it would be better to restrict to only the most relevant. Around 40% of the citations have one or more of the co-authors of the present study, this self-citation percentage is quite high so perhaps some of these references could be trimmed.

It would be helpful to have a figure to outline the protocol of the three experiments with aims for each. Or alternatively describe each experiment with a sentence in the Overview section of the Materials and Methods. This would be useful to introduce them, with more detail provided later.

Abstract

1. Line 39 - 41: this sentence could be rephrased for clarity.
2. Line 43: The last two sentences do not seem to reflect the section on "implications" in the discussion of the paper.
3. Line 52: A phrase that is more descriptive than "challenge inhibitory control" would help with understanding the methodology used in the study, this sentence could be rephrased for clarity.
4. Line 54: This study does have relevance for conditions where there are pathological shifts in monoamines, but the link to muscle cramps is a bit more speculative.

Introduction

5. Line 61: It would be helpful to have a more direct link to the referenced studies given within the sentence.
6. Line 98 - 100: very long sentence, could be broken up for readability.

Methods

7. Line 131: if after practice the participant displayed more than 5% error, was this trial discarded from further analysis?
8. Line 137: Is there a reason for plateau one? If so, it could be mentioned here. Could the same result be achieved from just doing an initial ramp to 30%MVT and then lowering to plateau two?
9. Is Experiment 1 the same as Experiment 3 except that it is performed with the ankle at 110 degrees and Experiment 3 uses 70 and 90 degrees? Could their description be merged?
10. Line 200: was the decomposition performed on the monopolar or differential EMG signals?
11. Line 203: was any particular software used to edit the motor unit spike trains? Can a reference be added here to describe the procedures for manual editing in more detail (the editing process is outlined briefly in the reference currently given, but it would be good if there was a more detailed reference available)?
12. Line 179: Here the per trial number of motor units would be more useful (mean +- standard deviation), and perhaps also the percentage used in the analysis from the number of motor units decomposed. If this different for the different types of trials, this could also be mentioned, and a separate mean and standard deviation reported for each.
13. How many motor units were used for the delta F calculation (average and standard deviation per trial) after satisfying all

the criteria?

14. Line 198: Motor unit decomposition is challenging when changing force levels. Can the authors comment on whether any specific method was used to decompose the trials (e.g. split into sections and merged)? Or was informed decomposition used where the motor units filters from the isometric contractions were used in the decomposition of the sombrero contractions?

15. Line 278: What significance threshold was used for the coherence? Is there a reason only common drive was included and not the other frequency bands?

Results

16. Line 345: Is the sombrero trajectory itself "constraining inhibitory control"? Perhaps the title could be something more intuitive like "examining prolongation and its effect on force using sombrero trajectories".

17. Line 466: As prolongation can also be observed using trapezoidal contractions, it might be good to emphasize the relative advantages of trapezoidal and sombrero type profiles. What is the advantage of using the sombrero for estimating prolongation?

18. Line 467: Can delta F be accurately estimated for an individual motor unit? Is it not more typical to use the average delta F within a trial? What was the inter-trial variability and inter-muscle variation in the delta F across participants?

19. Line 474: was the duration of firing averaged over all motor units for a given trial?

20. Line 452: Table 1: Some additions/changes would make the information presented here more interpretable. The number of trials, number of motor units decomposed per trial and number of unique motor units matched would be helpful. Also, if matching using multiple trials across conditions, it's possible that the same motor unit will be marked as a matching pair multiple times which would bias the results. Do the numbers indicate unique motor unit pairs?

21. Line 532: These results could be easier to present in a table.

Discussion

22. Line 570: To "impede motor control" might be too strong for the results shown.

23. Line 571-573: suggest rephrasing for clarity.

24. Line 586: introduced difficulties in control → increased the CV of the force produced.

25. Line 594: It may not be compensatory. Another simple factor in the increase in the coefficient of variation in plateau two is that more higher threshold motor units are active and higher threshold units tend to have larger coefficients of variation in their firing rates.

26. Line 599: The model results are very interesting, but to appreciate them and understand them I think they need to be discussed more. Also they are presented for what seems like the first time (if just looking at the main paper) on the first page of the discussion. They take up quite a prominent role considering they were only briefly mentioned in the results.

27. Line 607: In this case, it might be too much to say that the increase in variability is a "degradation in control", is this change in CV not just a consequence of the larger motor units remaining active at lower force levels?

28. Line 641: This section could alternatively belong in the limitations section.

29. Line 709: This paragraph could be reduced to 1-2 sentences, as it might be a bit speculative.

Figures

30. In all figures, it would be useful to have significance bars to make it clearer what differences/changes are of note.

31. Figure 1: Part (c) is not easy to interpret, perhaps there is an alternative way of presenting the information?

32. Figure 3: Could explicitly state that this figure is just brim MUs.

33. Figure 5: Overall, the message of Figure 5 is difficult to see because there is so much information presented. Is (a) across all subjects/trials? Part (b) is a bit unclear as there is a lot going on in this subplot, again there might also be a better way of representing averages than the triangles.

34. Figure 6. If presenting coherence, it would be good to also show example coherence spectra.

END OF COMMENTS

Referee #1:

This study examined a motor control paradox that may exist in persistent inward current function for motoneurons, where the known amplification of currents and prolongation of discharge may not be mutually beneficial for all motor control tasks. I read this manuscript with great interest, and I believe that the outcomes of the study will be of great interest to the wider motoneurone and physiology community. The team that performed this work has an excellent track record in the field and they have produced another excellent series of experiments. They present a novel contraction task referred to as 'sombbrero' contractions, which members of this team were the first to present in preprint (BioRxiv 2023) and was employed by the Kavanagh lab (J Physiol, 2024) and the Enoka lab (Exp Brain Res, 2024).

My major comment lies in the style of writing, as I think there needs to be changes made to the manuscript which remove the colloquial language entirely, and greater emphasise the physiology underpinning the experiments. For example, the introduction provides a good overview of background literature, as well as areas that require further research, but as a reader I could not identify what the current experiment was testing. There was no purpose or hypotheses to guide the reader into the experiments that have been designed for the current study. Hence, the majority of my comments are based on clarity rather than quality of results (which are excellent). Similarly, there are several interpretations of physiological mechanisms in the results which would be better suited to the discussion. I believe with some further work on the clarity in the body text of the manuscript this would be an excellent contribution to physiology literature.

We thank you for your review and constructive feedback. We have attempted to address each comment and concern as detailed below. We have also made changes throughout the manuscript to improve the writing style.

Abstract

It would be worth considering the balance of the abstract, as half of it is currently devoted to the background of PICs and brainstem neuromodulation, and only a sentence or two is devoted to the actual experiments and results.

Thank you, we have added another sentence to detail the results and restructured to reduce PIC background.

Introduction

Line 79: Can there be more explanation provided for boosting currents. Not really clear from the text. We have adjusted the wording here to indicate the that the additional current from PICs augments excitatory synaptic currents.

Lone 86: Do monoamines adapt excitability? What is normal motor repertoire?

We have adjusted this sentence for clarity.

Line 88: not clear what this sentence is saying. What is tension?

While the two sentences that follow this statement explain what we mean by tension, we recognize tension may be an ambivalent word. We have changed “tension” to “potential for conflict”

Line 98: not clear what the perceptual error and conflicting inputs relate to. Is this statement necessary for the current study?

We believe that drawing relations to adjacent areas of research is valuable for framing the work and making it potentially more accessible to a wider audience. We are happy to remove this sentence if you feel it is unnecessarily distracting, otherwise we have left it in.

Please include a purpose and hypotheses so the reader can fully appreciate what experiment design and methods should be used. Some of these details are in the first few lines of the methods, but they could be even more explicitly related to the content of the introduction.

Thank you, we have added a few paragraphs to the end of the introduction to highlight our hypotheses, findings, and implications.

Methods

Line 129: Needs to state how many trials/attempts were used to obtain MVC, as well as the method (I'm assuming peak torque for all valid attempts).

We have added a few sentences to better detail our methods for obtaining MVCs.

The z-coherence bands do not align with most other studies use. Importantly, the authors have selected cut-offs at 10 Hz, whereas nearly all other studies have the band to quantify coherence associated with 10 Hz (e.g. 5-15 Hz). Physiological tremor arises from afferent activity and oscillatory activity at the spinal cord level, and manifests at 10 Hz for healthy individuals. Thus, the cut-offs won't help in understanding a key neural contributor to force characteristics.

We have decided to remove the coherence analysis from the paper, as it was not essential to our hypotheses or interpretations and added unnecessary complexity to the methods section (as pointed out by reviewer 2).

Results

I strongly recommend removing the methods-based content from the results. The inclusion of methods not only doubles-up on the existing methods section, but also creates a very long results section to read. As one example, the very first paragraph of the results could have lines 347 to 356 deleted and the actual results would still remain.

There are several instances of interpretation in the results. I would recommend reporting statistically-driven results in this section, and move physiological interpretations to the discussion so that all interpretations can be viewed together.

I also recommend tightening up the language to avoid superlatives. For example, the terms stark and appreciated in line 498 "A stark increase in PIC prolongation behavior can be appreciated in Figure 5a". Many sentences like this can be edited throughout the manuscript. There are also many instances where 'degrade', or 'impact' or, 'good control' are used which distract from the excellent graphs and reporting of data.

Thank you, we have adjusted the results section to minimize methods, superlatives, and interpretation. We have left some methods and framing at the onset of each results section to orient the reader to the purpose of each experiment.

There is only coherence reported for the delta band and no mention of any other band? If this data has been analyzed as per the methods it is important to present it in results. I understand that delta coherence is the main determinant of force control during steady state contractions, but the other bands are still important for understanding how the CNS regulated motoneurone discharge.

We have removed this as detailed above.

Despite these comments, I must commend the authors on the quality of their graphs and illustrations. They are superb.

Thank you, we are glad that you find the graphs and illustrations informative.

Discussion

Once again I would recommend clear descriptions of findings. Statements such as 'we showcase', 'introduce impediments', 'engender aberrant discharge patterns', 'poorly controlled', 'challenging the inhibitory control' and 'problems in torque control' are not as informative as a direct description of the outcomes. These phrases are all in the first paragraph of the discussion. A few instances throughout the discussion are fine but not repeatedly.

Thank you. We have addressed all of the mentioned phrases and adjusted throughout.

Referee #2:

The overall finding reported in this manuscript is that the prolongation of motor unit discharges from the activation of PICs increases the coefficient of variation of force. This makes sense as larger motor units are still active at a lower force level than they would typically be recruited at, increasing the variability of the force. The secondary finding is that prolongation (and level of PICs), and thus CV of the force, is increased in conditions where there is likely to be less motor unit inhibition (shortened muscle lengths). This study shows that although PICs can aid in force production, they can also increase the variability of force produced. This has implications for conditions in which PICs are altered due to pathological shifts in monoamines.

These findings are easy to understand and explain in terms of the underlying physiology, but at the moment the main message of the paper is obscured by the number of results presented. Some restructuring of how the results are presented, and perhaps moving some supplementary analyses to the appendix/supplementary material, should help to distinguish the main message.

Thank you for your many comments and suggestions. We have responded to each of your comments below. Additionally, in an attempt to reduce the results presented, we have removed the coherence analysis as it was not a major theme.

Main comments:

1. At the moment it is difficult to see how the three experiments fit together from the introduction and methods, and the description of the experiments is a bit disjointed. Their relationship is clearer after reading the results, but it would be better to bring this in earlier in the paper. Some restructuring would be helpful so that the combined aim is more obvious earlier in the manuscript, but more importantly so that it is easy to see how each experiment contributes to this aim. In the introduction or methods it could be stated whether multiple experiments are being used to test the same hypothesis, or whether each tests a unique aspect of the overarching hypothesis.

Although the introduction provides a nice overview of PICs, more it could be dedicated to a specific discussion of the background needed to support the aims of this study. Although there is a hypothesis given, the overarching aim (and specific aims of each sub-experiment) could be briefly mentioned here. The methods could also be restructured to make the aims a bit clearer. One suggestion is to include a very brief outline of the methods used for each experiment in the first paragraph (mirroring the aims that could be included at the end of the introduction). The more detailed information can then be addressed later in the methods. Some signposting indicating which sections will outline methodology shared between the experiments and which is specific to a particular experiment could also be helpful.

Thank you for this suggestion. We have added three paragraphs to the introduction to better outline how each experiment and hypotheses fit within the overarching paper.

2. There are lots of different methods presented: sombrero profile, delta F, motor unit coherence, motor unit matching and model simulations. Dividing the focus of the paper over all of these different methodologies may be confusing the main message. Are all of these methodologies essential to the results of the paper and are they all needed in the main body of the paper? This may be the first time the sombrero contraction is presented (for peer reviewed research), so it would be good to make sure it is the focus.

Thank you for pointing this out. We have adjusted the paper slightly and removed the coherence results to reduce the number of analyses/methods, as the coherence was not a main theme in the manuscript.

3. Changing the muscle length would presumably alter the motor unit action potential waveform recorded at each electrode which would make matching based on action potential shapes very difficult to do accurately. This could be mentioned as a limitation. In this case, I think visual inspection of matched MUs is also needed for the challenging scenario of matching across different muscle lengths. Some examples of matched MUAP templates in the supplementary material could also help this. Furthermore, it's not clear from the methods if it would be possible to have matches that include the same motor unit (or pair of motor units) multiple times or repeated pairs/motor units are excluded.

Yes, we agree, matching with waveforms is not perfect in this scenario. We have added this as a limitation. Repeated pairs of motor units were concatenated when matched between successive joint angles; we have added a clarifying statement in the methods.

4. For the statistical analysis, if the number of motor units per trial differed significantly this could have a large effect on the results (as this number is being used to normalize the results). Another way of looking at this would be to use the number of motor units as a factor in the linear mixed model rather than using it to normalize the number of cap MUs. Does the result of a higher number of cap MUs at shorter muscle lengths still hold in this case? Also, is the delta F measure affected by the number of motor units decomposed? This is also something that could be investigated using the LMM.

Thank you for this comment and we appreciate the suggestion. The only values that are normalized by the number of cap MUs, are the proportion variables. All other values are *not* normalized by the total number of cap MUs. We agree, normalizing by cap MUs would be problematic, but since we are only doing this for a single outcome metric to create a proportion, we believe the remaining metrics should be fine. The results of a higher number of cap MUs at shorter muscle lengths is a strong results and is shown in Figure 5d bottom panel. The average number of cap MUs for each trial is shown in this plot.

Minor:

There are over 100 references, this number could be reduced. Some statements are supported by 6 references, it would be better to restrict to only the most relevant. Around 40% of the citations have one or more of the co-authors of the present study, this self-citation percentage is quite high so perhaps

some of these references could be trimmed.

Thank you for bringing this to our attention, we have trimmed our references, leaving those we see most useful for the reader.

It would be helpful to have a figure to outline the protocol of the three experiments with aims for each. Or alternatively describe each experiment with a sentence in the Overview section of the Materials and Methods. This would be useful to introduce them, with more detail provided later.

Thank you for this suggestion. We have added a paragraph in the methods overview to introduce each experiment.

Abstract

1. Line 39 - 41: this sentence could be rephrased for clarity.

We have revised this sentence for clarity.

2. Line 43: The last two sentences do not seem to reflect the section on "implications" in the discussion of the paper.

Thank you, we have added a sentence for potential implications.

3. Line 52: A phrase that is more descriptive than "challenge inhibitory control" would help with understanding the methodology used in the study, this sentence could be rephrased for clarity.

We have revised this sentence for clarity.

4. Line 54: This study does have relevance for conditions where there are pathological shifts in monoamines, but the link to muscle cramps is a bit more speculative.

Yes, we agree. We have adjusted to indicate it is entirely speculative.

Introduction

5. Line 61: It would be helpful to have a more direct link to the referenced studies given within the sentence.

We have removed this citation entirely. It is not strictly necessary, as this is mostly common knowledge.

6. Line 98 - 100: very long sentence, could be broken up for readability.

Thank you, we have adjusted this sentence.

Methods

7. Line 131: if after practice the participant displayed more than 5% error, was this trial discarded from further analysis?

Thank you, we have added the following sentence to clarify. "During each experiment, all trials were included unless the participant was unable to achieve the desired torque trace, ceased the trial prematurely, or substantially deviated from the torque trace $\pm 5\%$."

8. Line 137: Is there a reason for plateau one? If so, it could be mentioned here. Could the same result be achieved from just doing an initial ramp to 30%MVT and then lowering to plateau two?

Yes, we chose to include an initial plateau to observe how the lower threshold motor units adapted to the prolongation from the newly recruited higher threshold units (e.g., brim motor units; figure 3). Using the sombrero task is simply the easiest way to accomplish this.

9. Is Experiment 1 the same as Experiment 3 except that it is performed with the ankle at 110 degrees and Experiment 3 uses 70 and 90 degrees? Could their description be merged?

Yes, these experiments could conceivably be merged. We have tried to keep duplicative information to a minimum when explaining the methods and refer back to the prior experiments when relevant. We would prefer to keep the experiments separate to follow the intended flow of the manuscript.

10. Line 200: was the decomposition performed on the monopolar or differential EMG signals?

Differential

11. Line 203: was any particular software used to edit the motor unit spike trains? Can a reference be added here to describe the procedures for manual editing in more detail (the editing process is outlined briefly in the reference currently given, but it would be good if there was a more detailed reference available)?

Custom software was used. We have added two references, Del Vecchio 2020 overviewing the general process and Hug 2021 detailing the reliability of manual editing.

12. Line 179: Here the per trial number of motor units would be more useful (mean \pm standard deviation), and perhaps also the percentage used in the analysis from the number of motor units decomposed. If this different for the different types of trials, this could also be mentioned, and a separate mean and standard deviation reported for each.

We have removed this from the methods, as it is more relevant in the results section, and added mean \pm std in table 1.

13. How many motor units were used for the delta F calculation (average and standard deviation per trial) after satisfying all the criteria?

The average number of decomposed units for each trial can be seen in table 1. For the ΔF calculation, all MUs with a recruitment threshold lower than itself were used as reporter units. That means that lower threshold units necessarily had fewer reporter units than higher threshold units. ΔF for each unit is then

the average of all of these possible pairs. Our interest in using ΔF in this paper was not to make an argument on the validity or methodology of ΔF . We are using standard approaches and treating ΔF as a “gold standard” in experiment 2 to ensure that the intended changes in excitability are achieved with muscle length. We wish to save any speculation on the validity of ΔF and potential alternative approaches for manuscripts where that is of greater focus.

14. Line 198: Motor unit decomposition is challenging when changing force levels. Can the authors comment on whether any specific method was used to decompose the trials (e.g. split into sections and merged)? Or was informed decomposition used where the motor units filters from the isometric contractions were used in the decomposition of the sombrero contractions?

No such strategies were performed at the time, though we acknowledge that this could have presumably increased yield.

15. Line 278: What significance threshold was used for the coherence? Is there a reason only common drive was included and not the other frequency bands?

Yes, we never intended for the coherence analysis to be a main theme within the paper and thus wanted to keep the reporting/analysis minimal. As indicated above, we have chosen to remove the coherence analysis.

Results

16. Line 345: Is the sombrero trajectory itself "constraining inhibitory control"? Perhaps the title could be something more intuitive like "examining prolongation and its effect on force using sombrero trajectories".

Thank you, we have used your suggested title.

17. Line 466: As prolongation can also be observed using trapezoidal contractions, it might be good to emphasize the relative advantages of trapezoidal and sombrero type profiles. What is the advantage of using the sombrero for estimating prolongation?

Thank you, we have added a paragraph in the discussion to highlight the advantages of the sombrero.

18. Line 467: Can ΔF be accurately estimated for an individual motor unit? Is it not more typical to use the average ΔF within a trial? What was the inter-trial variability and inter-muscle variation in the ΔF across participants?

ΔF is a pool-level metric, and we generally treat it as such. Averaging values across trials would eliminate variability in its estimation, artificially inflating confidence in differences between conditions and failing to capture the true variability of the data. To address this, we employed linear mixed modeling methods, clustering within the relevant hierarchical structure. This approach accounts for variability while allowing for accurate estimation of differences between conditions.

19. Line 474: was the duration of firing averaged over all motor units for a given trial?

No, it was not.

20. Line 452: Table 1: Some additions/changes would make the information presented here more interpretable. The number of trials, number of motor units decomposed per trial and number of unique motor units matched would be helpful. Also, if matching using multiple trials across conditions, it's possible that the same motor unit will be marked as a matching pair multiple times which would bias the results. Do the numbers indicate unique motor unit pairs?

We have updated the table to indicate the average MUs per trial. We attempted to match and identify unique motor units across all possible pairs. When a match was found, the units in the pair were assigned a unique motor unit ID (MUID). Any pairs that shared matches were grouped under the same MUID. For example, if MU-A matched MU-B, MU-C matched MU-D, and MU-B matched MU-D, all four (MU-A, B, C, and D) were assigned the same MUID. This MUID was then used in the mixed model as described in the statistical analysis section. While we are confident this procedure minimizes duplicate pairs, there still remains a small possibility that matches are missed

21. Line 532: These results could be easier to present in a table.

Thank you, we do not wish to add any further tables to the manuscript at this point, unless strictly necessary.

Discussion

22. Line 570: To "impede motor control" might be too strong for the results shown.

We agree and have adjusted.

23. Line 571-573: suggest rephrasing for clarity.

We have adjusted the introductory paragraph for clarity.

24. Line 586: introduced difficulties in control → increased the CV of the force produced.

We have changed to torque CV.

25. Line 594: It may not be compensatory. Another simple factor in the increase in the coefficient of variation in plateau two is that more higher threshold motor units are active and higher threshold units tend to have larger coefficients of variation in their firing rates.

We agree and have removed the language suggesting a compensatory strategy.

26. Line 599: The model results are very interesting, but to appreciate them and understand them I think they need to be discussed more. Also they are presented for what seems like the first time (if just looking at the main paper) on the first page of the discussion. They take up quite a prominent role considering they were only briefly mentioned in the results.

The simulation results are meant to be a very brief addition to support speculative discussion. We do not wish to amplify the role of the simulations in this manuscript given the already burdensome length of the results presented. They do not serve any strong purpose in the manuscript and are in the supplementary sections for the interested reader.

27. Line 607: In this case, it might be too much to say that the increase in variability is a "degradation in control", is this change in CV not just a consequence of the larger motor units remaining active at lower force levels?

While we appreciate the nuance, we see this a degradation in control. We see being unable to derecruit higher threshold MUs and thus unable to control torque as precisely as once possible as an issue in the control of motor output.

28. Line 641: This section could alternatively belong in the limitations section.

We see changes in Ib feedback as a potential alternative interpretation and not a limitation.

29. Line 709: This paragraph could be reduced to 1-2 sentences, as it might be a bit speculative.

We have shorted this paragraph and highlighted that it is entirely speculative.

Figures

30. In all figures, it would be useful to have significance bars to make it clearer what differences/changes are of note.

Thank you. In figures 2 and 3 we have added pointers to indicate the significance associated with the Cohens d values on the right most column. Figure 4 has significance bars at the bottom of the plot. In figure 5, we have indicated significance with the Cohen's d values and colored bars pointing out which conditions are indicated. We have added clarifying statements indicating only significant differences are represented with Cohen's d values. We apologize, we thought we had originally included such a statement.

31. Figure 1: Part (c) is not easy to interpret, perhaps there is an alternative way of presenting the information?

Thank you for this suggestion; we appreciate it. While we have contemplated many ways to display the information, we have decided to remain with the current form.

32. Figure 3: Could explicitly state that this figure is just brim MUs.

Thank you, we have explicitly stated them as brim units.

33. Figure 5: Overall, the message of Figure 5 is difficult to see because there is so much information presented. Is (a) across all subjects/trials? Part (b) is a bit unclear as there is a lot going on in this subplot, again there might also be a better way of representing averages than the triangles.

Yes, (a) represents the entire population, as indicated in the figure caption. We prefer to present all the data whenever possible. While we acknowledge that the figures contain a lot of detail, we believe the current design is the clearest. We explored several adjustments to (b) but found the current format to be the simplest. Coloring the red dots by participant, for example, proved too distracting. Removing the raw data and displaying only participant averages is not ideal, as it limits the reader's ability to fully appreciate the dataset. Alternatively, replacing participant averages with model estimates would omit valuable information about individual variability. The inclusion of triangles provides a balanced solution, offering participant means without detracting from the primary narrative.

Dear Professor Heckman,

Re: JP-RP-2025-287446R1 "Intrinsic properties of spinal motoneurons degrade ankle torque control in humans" by James A Beauchamp, Gregory EP Pearcey, Obaid U Khurram, Francesco Negro, Julius P.A. Dewald, and C. J. Heckman

We are pleased to tell you that your paper has been accepted for publication in The Journal of Physiology.

- Your paper contains Supporting Information of a type that we no longer publish, including supplementary tables and figures. Any information essential to an understanding of the paper must be included as part of the main manuscript and figures. The only Supporting Information that we publish are video and audio, 3D structures, program codes and large data files. Your revised paper will be returned to you if it does not adhere to our Supporting Information Guidelines.

You are welcome to provide this material within the article, as an Appendix, instead.-

Yours sincerely,

Richard Carson
Senior Editor
The Journal of Physiology

If you would like to receive our 'Research Roundup', a monthly newsletter highlighting the cutting-edge research published in The Physiological Society's family of journals (The Journal of Physiology, Experimental Physiology, Physiological Reports, The Journal of Nutritional Physiology and The Journal of Precision Medicine: Health and Disease), please click this link, fill in your name and email address and select 'Research Roundup':

<https://www.physoc.org/journals-and-media/membernews>

- **TRANSPARENT PEER REVIEW POLICY:** To improve the transparency of its peer review process, The Journal of Physiology publishes online as supporting information the peer review history of all articles accepted for publication. Readers will have access to decision letters, including Editors' comments and referee reports, for each version of the manuscript as well as any author responses to peer review comments. Referees can decide whether or not they wish to be named on the peer review history document.
- You can help your research get the attention it deserves! Check out Wiley's free Promotion Guide for best-practice recommendations for promoting your work at: www.wileyauthors.com/eoo/guide. You can learn more about Wiley Editing Services which offers professional video, design, and writing services to create shareable video abstracts, infographics, conference posters, lay summaries, and research news stories for your research at: www.wileyauthors.com/eoo/promotion.
- **IMPORTANT NOTICE ABOUT OPEN ACCESS:** To assist authors whose funding agencies mandate public access to published research findings sooner than 12 months after publication, The Journal of Physiology allows authors to pay an Open Access (OA) fee to have their papers made freely available immediately on publication.

EDITOR COMMENTS

Reviewing Editor:

All of the reviewers' comments have been satisfactorily addressed. Reviewer 2 has a small number of minor suggestions that may be helpful in preparing the final version of the manuscript for publication.

Senior Editor:

Please see the comments provided by Referee #2 and by the Reviewing Editor concerning minor changes that are requested when preparing the final copy of the manuscript.

REFEREE COMMENTS

Referee #1:

I have reviewed the revised manuscript and the changes that the authors have made has increased the clarity and strengthened the outcomes of the study. The authors have produced an excellent study.

Referee #2:

The authors have successfully addressed my comments. A reduction in the number of results presented has made the manuscript much clearer and highlights the novelty of the study. The introduction reads very well now and the rationale for the study is clearly outlined.

I have a few minor suggestions for improvements:

Line 35: "Here, we designed two motor tasks that challenged the inhibitory control of PICs, generating unduly PIC prolongation and quantifiable deficits in human torque control." It is a minor suggestion on language, but as this is an important introductory sentence it would be good to be more precise in what "unduly" and "quantifiable deficits" mean. Perhaps "undesirable prolongation that decreased torque stability" or something similar.

Line 277: "To improve decomposition accuracy, automatic decomposition results were augmented by iteratively re-estimating the spike train and correcting for missed spikes or substantial deviations in the discharge profile (Del Vecchio et al., 2020; Hug et al., 2021; Martinez-Valdes & Negro, 2023)." - In the rebuttal the authors mentioned that manual editing was used, but this line is a bit ambiguous and could also be interpreted as implying automatic adjustment/editing was used. It would be helpful to explicitly state that manual editing of the spike trains was performed.

Line 645: "less they engender aberrant discharge patterns" could be phrased a bit more simply.

Line 677: "the presence of cap MUs could put motor output in a higher gain state" - this is an interesting hypothesis, is there anything further that the authors could add as to why the presence of cap MUs could alter the gain?

I appreciate that the authors have tried to think of alternative ways to present Figure 1 (c), but a few small changes might make a big difference in how clear the results are. There's a lot of overlap in the data points, both the triangles and the circles, that obscures the values. Adding horizontal offsets to the triangles and making the circles dots (with jitter so you can see the individual data points) would help. Also up to the authors' choice, but for completeness it would also make sense to include the average of the button MUs for each subject, in the same colour, but it may mean splitting the figure into two separate subplots for readability.

With the authors' recent studies on the differences in PICs between males and females, it would also be worth mentioning that there was the majority of the participants recruited in the study were male (10 out of 12 participants).